# A DNA origami-based aptamer nanoarray for potent and reversible anticoagulation in hemodialysis

Shuai Zhao[1,2,3,7], Run Tian[1,2,7], Jun Wu[4,7], Shaoli Liu[1,2], Yuanning Wang[1], Meng Wen[4], Yingxu Shang[1,2], Qing Liu[1], Yan Li[1], Ying Guo[5], Zhaoran Wang[1,2], Ting Wang[1], Yujing Zhao[4], Huiru Zhao[4], Hui Cao[4], Yu Su[4], Jiashu Sun [1,2], Qiao Jiang [1,2 ✉] & Baoquan Ding [1,2,6 ✉]

Effective and safe hemodialysis is essential for patients with acute kidney injury and chronic renal failures. However, the development of effective anticoagulant agents with safe antidotes for use during hemodialysis has proven challenging. Here, we describe DNA origami-based assemblies that enable the inhibition of thrombin activity and thrombus formation. Two different thrombin-binding aptamers decorated DNA origami initiates protein recognition and inhibition, exhibiting enhanced anticoagulation in human plasma, fresh whole blood and a murine model. In a dialyzer-containing extracorporeal circuit that mimicked clinical hemodialysis, the origami-based aptamer nanoarray effectively prevented thrombosis formation. Oligonucleotides containing sequences complementary to the thrombin-binding aptamers can efficiently neutralize the anticoagulant effects. The nanoarray is safe and immunologically inert in healthy mice, eliciting no detectable changes in liver and kidney functions or serum cytokine concentration. This DNA origami-based nanoagent represents a promising anticoagulant platform for the hemodialysis treatment of renal diseases.

[1] CAS Key Laboratory of Nanosystem and Hierarchical Fabrication, CAS Center for Excellence in Nanoscience, National Center for Nanoscience and Technology, 100190 Beijing, China. [2] University of Chinese Academy of Sciences, 100049 Beijing, China. [3] Sino-Danish College, Sino-Danish Center for Education and Research, University of Chinese Academy of Sciences, 100049 Beijing, China. [4] Beijing Jishuitan Hospital, Department of Laboratory Medicine, Peking University Fourth School of Clinical Medicine, 100035 Beijing, China. [5] National & Local Joint Engineering Research Center of Biodiagnosis and Biotherapy, The Second Affiliated Hospital of Xi'an Jiaotong University, 710004 Xi'an, China. [6] School of Materials Science and Engineering, Zhengzhou University, 450001 Zhengzhou, China. [7] These authors contributed equally: Shuai Zhao, Run Tian, Jun Wu. ✉email: jiangq@nanoctr.cn; dingbq@nanoctr.cn

Hemodialysis is a standard extracorporeal procedure for patients with chronic renal failure to remove waste products from the blood. During the treatment, patients' blood is cleansed by filtration through a dialysis circuit and artificial kidney (dialyzer), a process that has the tendency to activate platelets and coagulation cascades. Prevention of clotting in the extracorporeal circuit is one of the major challenges in safe and effective hemodialysis treatment. During clinical hemodialysis procedures, several anticoagulant drugs, including unfractionated heparin (UFH) and low molecular weight heparin (LMWH), are frequently administered, enabling robust anticoagulation by accelerating antithrombin III (ATIII)-mediated irreversible inhibition of multiple procoagulant proteases[1–3]. Although widely used, these heparin-like compounds still have a range of undesirable side effects, including thrombocytopaenia (HIT) and possible immune-mediated disorders, which hamper their application in hemodialysis[2–4]. Moreover, protamine, the clinically used antidote of UFH, has been reported to elicit toxicity through several different mechanisms; protamine partially reverses the effect of ATIII-dependent LMWH, such as enoxaparin, but has no corrective activity on shorter heparins[5]. Because of the lack of safe and effective antidotes to reverse the anticoagulation of heparin-like compounds, post-treatment bleeding remains a major adverse event for hemodialysis patients. Thus, a strong clinical need remains for the development of effective and antidote-controllable alternatives to heparin-like compounds.

Thrombin is a vital enzyme in the coagulation system and an attractive target for the design of anticoagulant drugs[6–9]. A series of thrombin-binding aptamers have been screened by systematic evolution of ligands by exponential enrichment (SELEX), an in vitro selection technique[10]. These aptamers can directly recognize and inhibit thrombin, while the binding to thrombin can be ceased through the hybridization of complementary oligonucleotides (antidotes)[11–13], making these aptamers promising anticoagulant drug candidates. However, naked or unmodified aptamers suffer from inefficient delivery, fast degradation and rapid renal clearance in vivo, severely limiting their application in anticoagulation[14,15]. Although several nanoparticle (NP)-based strategies have been reported to enhance the stability and bioavailability of thrombin-binding aptamers[16–18], those approaches do not offer precise control of the density and orientation of aptamers displayed on the surface of NPs. These characteristics of the delivery vectors are important to affect efficient thrombin inhibitory activity. In contrast to traditional nanoparticles, DNA/RNA nanotechnology offers a variety of nanostructures with rationally designed geometric features, precise spatial addressability and remarkable biocompatibility[19–24]. Several aptamer-tagged DNA/RNA nanostructures have been utilized for immobilizing proteins with defined nanometer spacing and precision[23–26]. Though the reported nanoarchitectures have achieved substantial anticoagulant effects in plasma in test tubes[24,26], in vivo study and clinically relevant validation have yet to be reported.

In this work, we describe a supramolecular approach to construct a nanoscale anticoagulant agent based on self-assembled DNA origami. We generate DNA origami as a template to incorporate thrombin-binding aptamers with precise nanometer spatial control (Fig. 1). Through optimizing the types, distances and numbers of aptamers, we demonstrate that DNA origami-based aptamer assemblies (Aptarrays) can specifically and efficiently bind thrombin molecules, inhibiting their coagulation function (Fig. 1). Importantly, Aptarray effectively exert anticoagulation in human plasma, fresh whole blood, a murine model and a dialyzer-containing extracorporeal circuit that mimics clinical hemodialysis. We also design antidote nucleotides that promptly reverse the anticoagulant effects engendered by the DNA nano-anticoagulants.

## Results

**Construction of DNA origami-based bi-aptamer arrays.** Based on Rothemund's method[19], rectangular DNA origami templates ($90 \times 60 \times 2$ nm) were assembled by slow annealing M13 bacteriophage genome DNA strands, multiple staple strands and capture strands at a molar ratio of 1:5:5 from 95 °C to room temperature (Fig. 1a). Assembled origami templates were purified using filtration devices (100 kDa MWCO, Amicon, Millipore) to remove the excess short-DNA strands. Two groups of capture strands (denoted blue and red in Fig. 1a and Supplementary Figs. 1–7) were extended from the addressable surfaces of the templates and functioned as binding sites to assemble two types of thrombin-binding aptamers. TBA15 is a 15-base-long, single-stranded DNA oligonucleotide that can directly bind to exosite I (the fibrinogen recognition exosite) of thrombin (Kd ~70–100 nM)[27] and elicit a potent anticoagulant activity. Another antithrombin aptamer, HD22, recognizes thrombin's exosite II (the heparin-binding exosite) with a high binding affinity (Kd ~0.5 nM)[28]. The DNA origami technique allowed us to pin the capture strands in the desired position of the addressable template surface, engineering the functional DNA assemblies with precisely controlled binding sites for the two thrombin-binding aptamers. Two different capture sequences were used for the two aptamers to avoid nonspecific binding. In our design, two rows of nine of each binding site were arranged ~68, 46, 24, or 5.4 nm apart, respectively (see Supplementary Figs. 1–4 for the detailed design of nanostructures I–IV). After annealing them by decreasing the temperature from 45 °C to 25 °C for six cycles, we analyzed these four aptamer-loaded assemblies by atomic force microscopy (AFM, Supplementary Fig. 8). The extended aptamers on the DNA origami were able to recognize thrombin molecules with dimensions of ~4 nm[29], allowing them to be anchored on the surface of the DNA sheet. Thrombin-binding assays demonstrated that optimal protein recognition was obtained from the DNA origami-bivalent aptamer assemblies with distances of 5.4 nm (Supplementary Figs. 9–13). As the inter-aptamer distance of 5.4 nm matches the dimension of the thrombin molecule (~4 nm), the two aptamers can act as bivalent single molecular species that display a stronger binding affinity to the protein than any one of the individual aptamers does alone. The avidity effect of DNA origami-bivalent aptamer assemblies is consistent with the previously reported results[23,26]. Therefore, we designed and fabricated bi-aptamer arrays (36 TBA15 and 36 HD22, Aptarray) on DNA origami with inter-aptamer distances of 5.4 nm (Supplementary Fig. 14) for efficient thrombin binding.

The Aptarrays were subsequently purified by poly (ethylene glycol) (PEG)-induced precipitation to remove redundant aptamer strands[30] (Supplementary Fig. 15). The purified Aptarrays were then analyzed by agarose gel electrophoresis (Fig. 2a and Supplementary Fig. 16). Under ultraviolet (UV) illumination, the M13 DNA scaffold, DNA rectangular origami and three types of origami-aptamer assemblies (Ori-TBA15, Ori-HD22 and Aptarray) were apparent as distinct bands (Fig. 2a). After hybridization with Cy5-TBA15 and/or Alexa488-HD22, the DNA nanostructures bands in the agarose gels were observed via the Cy5 and/or Alexa488 excitation channel(s). The origami comigrated with Cy5-TBA15 and Alexa488-HD22, indicating that both aptamers were successfully attached to DNA origami template. TBA15 or HD22 aptamers were labeled with Alexa488 for quantification, and the concentration of TBA15 or HD22 on DNA origami was determined by measuring the absorbance of Alexa488 at 488 nm (Supplementary Fig. 17). The average

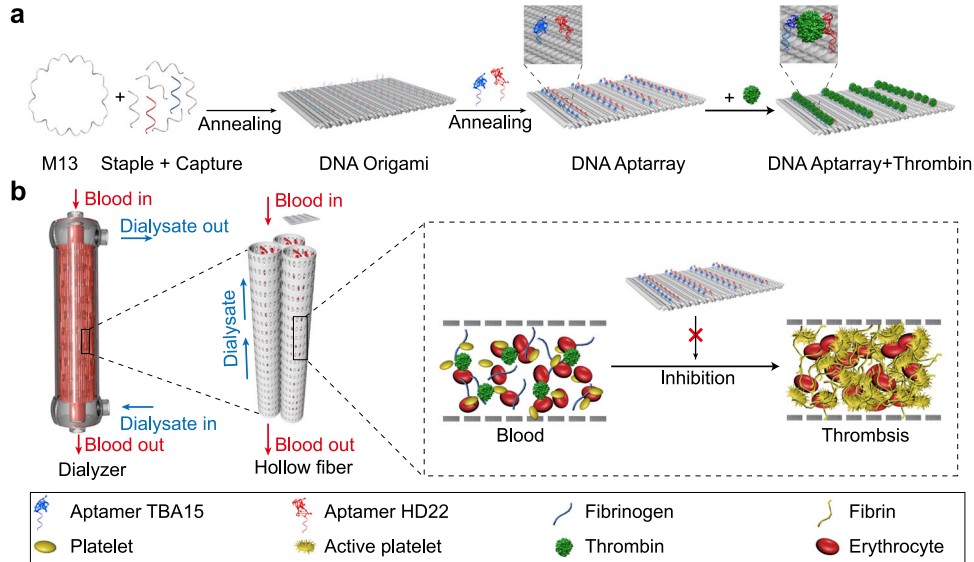

**Fig. 1 Design of an aptamer-functionalized DNA origami nanoarray for anticoagulation in hemodialysis. a** Schematic illustration of the construction of the aptamer-loaded nanoarray by DNA origami, and the specific binding and inhibition of thrombin. Single-stranded M13mp18 DNA (M13) is folded by annealing with staple strands and capture strands (staple + capture) to form a rectangular DNA origami. Two types of thrombin-binding aptamers, TBA15 and HD22, are loaded onto the DNA origami by hybridization with capture sequences (blue and red) that extend from the surface of the rectangular DNA template (DNA Aptarray). The Aptarray can specifically and efficiently bind thrombin molecules, inhibiting coagulation. **b** Schematic representation of the utilization of the DNA Aptarray for efficient anticoagulation in hemodialysis. Fresh human blood from healthy donors along with the Aptarray is pumped via mechanical roller pump to a dialysis column (Dialyzer) through the closed extracorporeal circuit, while the dialysate flows through the device and then discarded. Aptarray captures thrombin molecules and efficiently inhibits their catalytic activities, eliciting robust anticoagulatory effects in the ex vivo extracorporeal hemodialysis circuit.

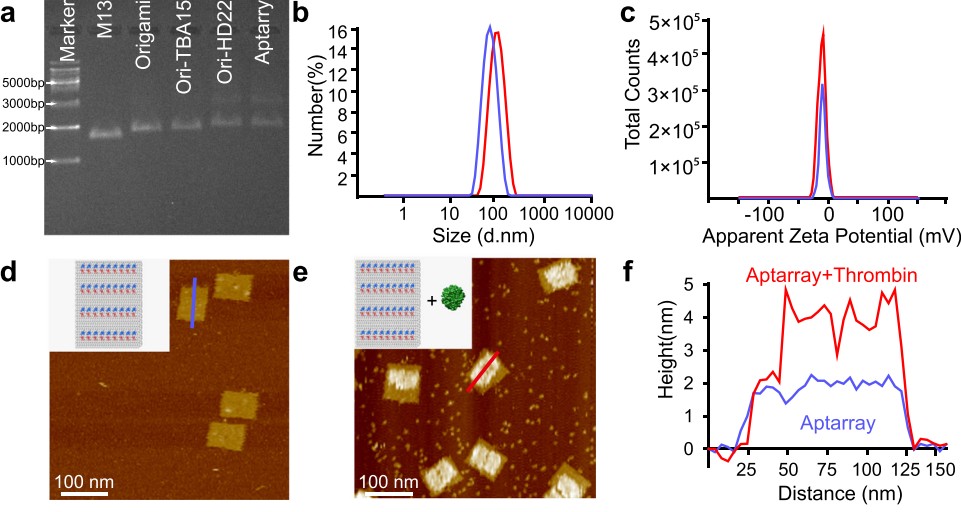

**Fig. 2 Characterization of DNA origami-based aptamer nanoarray. a** Ethidium bromide-stained agarose gel image of the step-by-step assembly of DNA origami-aptamer nanoarray. Lane 1: Marker. Lane 2: M13 genome DNA (M13). Lane 3: rectangular DNA origami template with capture strands (Origami). Lane 4: DNA origami-TBA15 nanostructures (Ori-TBA15). Lane 5: DNA origami-HD22 nanostructures (Ori-HD22). Lane 6: DNA origami-aptamer nanoarray (Aptarray). **b** Hydrodynamic sizes and **c** Zeta potentials of DNA origami template (blue) and Aptarray (red). **d** AFM image of bivalent aptamer arrays on the DNA origami (36 TBA15, blue; 36 HD22, red; the four double rows of aptamers with distances of 5.4 nm between aptamer types). **e** Thrombin incubation with the Aptarray illustrated in panel **d**. The large areas of bright spots on the surface of the rectangular origami represent the bound thrombin molecules. Scale bars, 100 nm. **f** The heights of origami-based aptamer nanoarrays (blue, Aptarray; panel **d**) and thrombin bound structures (red, Aptarray + Thrombin; panel **e**) were measured by AFM. The gel image (**a**) and AFM graphs (**d**, **e**) are representative of three independent experiments. Source data are provided as a Source Data file.

number of TBA15 and HD22 on each DNA origami sheet was calculated to be 33.5 ± 0.4 and 32.4 ± 1.1, respectively. The size and the zeta potentials of nanoparticles are important parameters to affect their behaviors in vitro and in vivo. Dynamic light scattering (DLS) analysis showed that the average diameters of

these DNA nanostructures increased from 63.0 ± 1.2 nm (bare origami, blue) to 92.2 ± 1.3 nm (Aptarray, red) after the hybridization of the two aptamers (Fig. 2b). The zeta potentials of DNA origami and Aptarray were −12.2 ± 0.8 mV and −10.9 ± 1.1 mV, respectively (Fig. 2c). AFM images show the successful

binding of thrombin molecules on the surface of Aptarray (Fig. 2d–f and Supplementary Fig. 18).

**Inhibition of thrombin catalytic activity.** To assess whether nanostructures with different distances between HD22 and TBA15 aptamers (~68, 46, 24, and 5.4 nm) could affect thrombin inhibition, we examined the effects of the origami-based nanostructures (I–IV) using clotting reactions to monitor the conversion of fibrinogen into fibrin. Thrombin can catalyze the conversion of soluble plasma protein fibrinogen into insoluble polymeric fibrin, leading to significant enhancement of a light scattering signal (Fig. 3a). Once aptamer pairs on the DNA origami capture thrombin molecules, the catalytic conversion of fibrinogen is expected to be inhibited. We investigated the clotting reaction using mixtures that contained thrombin, fibrinogen and the different nanostructures. Through light scattering measurements, the optimal inhibitory performance was observed by the DNA origami-bivalent aptamer nanostructures IV (with distances of 5.4 nm), in which the two aptamers elicited a potent synergistic effect on thrombin recognition and inhibition (Supplementary Fig. 19).

We next used the Aptarray (36 bivalent aptamer pairs with distances of 5.4 nm) for the subsequent anticoagulant experiments. The catalytic rate of thrombin ($V_{Cat}$) was partially inhibited by TBA15 or HD22 (20 nM; Supplementary Fig. 20)[31]. In comparison with the individual free aptamer strands, the mixtures of the aptamers and rectangular origami templates at equivalent aptamer amounts (Ori + T + H) resulted in slower fibrin

formation velocities (Fig. 3b, c). Other mixture controls (one type of aptamers is loaded on the origami and another type is mixed up in the solution, Supplementary Fig. 20) showed similar slow reaction curves. Aptamers tethered with 5-mer or 16-mer polyA linkers (T-5A-H, T-16A-H) induced slower fibrin formation velocities, in comparison with those control groups. Thrombin inhibitory performance can be significantly increased (98.3 ± 0.2%) by arraying an equivalent amount of aptamers on the DNA origami template with bi-aptamer distances of 5.4 nm (Aptarray: ~0.56 nM origami, containing 20 nM TBA15, 20 nM HD22, Fig. 3b, c). The Aptarray-induced inhibition is dose-dependent (Supplementary Fig. 21). These results demonstrated that DNA origami guided bi-aptamer array elicited a potent synergistic effect on thrombin recognition and inhibition. The effect of precise assembly induced-multivalency of bi-aptamers is important for thrombin inhibition. For different origami-based bi-aptamer structures with discrete numbers and controlled patterns, the results of the catalytic rate of thrombin revealed that nanoarrays with an equivalent bi-aptamers amount (20 nM TBA15, 20 nM HD22) exhibited a similar potent inhibition of thrombin (Supplementary Figs. 5–7 and 22). In comparison with T-16A-H, significantly greater inhibitory activities of the clotting reaction observed from the bi-aptamer nanostructures might be due to the higher scaffold rigidity, stronger electronegativity of DNA origami templates and more control on the orientation of bi-aptamer pairs[26,32].

We next measured thrombin inhibition in reaction mixtures that contained Aptarray, thrombin and fibrinogen after a 12 h incubation with fetal bovine serum (FBS). In this case, treatment

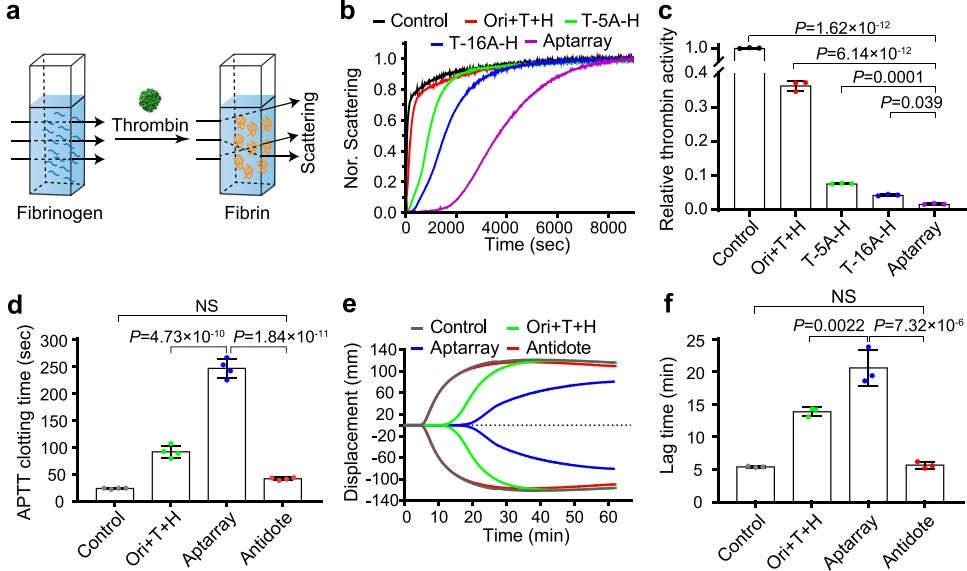

**Fig. 3 Inhibitory effects of the DNA origami-based aptamer nanoarray and anticoagulant reversal. a** Schematic drawing of scattering spectra detection of fibrin formation after the activation of fibrinogen by thrombin. **b** Light scattering spectra ($\lambda_{sc}$ = 650 nm) of a fibrinogen solution with thrombin only (Control) or thrombin with the mixture of origami and two types of aptamers (Ori + T + H: ~0.56 nM origami, 20 nM TBA15, 20 nM HD22), aptamers tethered with 5-mer (T-5A-H, 20 nM) or 16-mer polyA linkers (T-16A-H, 20 nM), or the DNA origami-aptamer nanoarray (Aptarray: ~0.56 nM origami, 20 nM TBA15, 20 nM HD22). **c** Relative thrombin activities of different treatments were estimated by the catalytic rate of thrombin with indicated groups. The catalytic rate of thrombin ($V_{cat}$) after the indicated treatments was calculated as $V_{Cat} = C_{Fibrinogen}/(t_{1/2} \times C_{Thrombin})$ obtained from the panel **b**. The concentration of fibrinogen ($C_{Fibrinogen}$) is 1 mg/ml, and the concentration of thrombin ($C_{Thrombin}$) is 12 nM. **d** Plasma clotting time (APTT) in the absence (Control) and presence of the mixture of origami and two types of aptamers (Ori + T + H: ~56 nM origami, 2 μM TBA15, 2 μM HD22), the DNA origami-aptamer nanoarray (Aptarray, ~56 nM origami, 2 μM TBA15, 2 μM HD22), and fivefold antidote-neutralized nanoarray (Antidote, 10 μM cTBA15 + cHD22, mixed with Aptarray for 5 min incubation at 25 °C). **e** Representative TEG tracing and **f** the time until detectable clot formation following kaolin-initiated coagulation for each condition described. The TEG assays were performed on human whole blood from healthy donors containing no anticoagulants (Control), one of the anticoagulant strategies (Ori + T + H or Aptarray) or neutralized anticoagulant (Antidote), with the maximum time limit of a TEG assay being ~60 min. Data represent the mean ± s.d. from either three (**c**, **f**) or four (**d**) independent replicates. Statistical significance (**c**, **d**, **f**) was calculated by one-way ANOVA with the Tukey post hoc test. NS, P > 0.05. Source data are provided as a Source Data file.

with the mixtures of aptamers and rectangular origami templates (Ori + T + H) after serum incubation produced only ~20% inhibition of thrombin function. In contrast, even after the 12 h treatment with serum, a significantly greater inhibition of thrombin activity (~82%) was observed with the Aptarray (Supplementary Fig. 23); the precisely organized aptamer array on the DNA origami template afforded a resistance against nucleases.

The inhibitory effects on thrombin induced by the Aptarray can be neutralized by the addition of sequence-specific complementary oligonucleotides (antidotes) that disrupt the aptamer secondary structures. We used mixtures with sequences complementary to thrombin-binding aptamers (cTBA15 + cHD22) to reverse the anticoagulant effect produced by treatment with the DNA Aptarray (Supplementary Fig. 24). The clotting reaction results demonstrate that treatment with an antidote mixture equal to that of the aptamers (onefold; cTBA15, 20 nM; cHD22, 20 nM) partially restored thrombin function (~20%). Three- or fivefold antidotes produced ~60% or ~80% of the thrombin activity to levels observed in the absence of anticoagulants.

**Anticoagulant activity in human plasma and whole blood.** We next investigated the anticoagulant activity of the Aptarray in human plasma. We compared the inhibitory efficacy of equivalent aptamer amounts of free aptamers (TBA15 or HD22), mixtures of aptamers and rectangular origami templates (Ori + T + H) and Aptarray using activated partial thromboplastin time (APTT; Fig. 3d and Supplementary Fig. 25) and prothrombin time (PT; Supplementary Fig. 25). The APTT and PT are commonly used medical tests that assess a person's blood coagulation process and monitor anticoagulation therapy. These assays evaluate the amount and the function of clotting factors that are important for blood clot formation[33]. The APTT examines the activity of the intrinsic and common clotting pathways, while the PT assay measures the integrity of the extrinsic and common systems of coagulation. Both tests reflect thrombin function and subsequent formation of the fibrin clot and are widely used for clinical investigation of clotting or bleeding disorders. Negative controls were prepared using buffer only in plasma from healthy donors. The control showed an average clotting time (APTT) of $24 \pm 1.1$ s (Fig. 3d). The presence of mixtures of aptamers and origami templates (Ori + T + H: ~56 nM origami, 2 μM TBA15, 2 μM HD22) resulted modest prolongation of clotting time ($92 \pm 11.4$ s), compared to the control group. The largest delay in clotting time, $246.7 \pm 17.5$ s, was observed by addition of Aptarray (Fig. 3d, Aptarray v.s. Ori + T + H, $P < 4.73 \times 10^{-10}$, calculated by one-way analysis of variance (ANOVA) with the Tukey post hoc test). The presence of the Aptarray resulted in a concentration-dependent effect in the APTT and PT assays (Supplementary Fig. 25). After addition of fivefold DNA antidote for a 5 min incubation at 25 °C, ~92% of the anticoagulant activity was reversed (Fig. 3d).

We next evaluated the anticoagulation ability of the Aptarray in whole blood. We first investigated the stability of the Aptarray in fresh whole blood using a hemolysis assay (Supplementary Fig. 26). Blood samples were treated with Triton X100, a detergent that permeabilizes cell membranes, exhibited strong hemolysis. In contrast to this positive control (Triton X100), buffer, mixture of the two aptamers (T + H, 2 μM TBA15 + 2 μM HD22), a mixture of DNA origami and aptamers (Ori + T + H: ~56 nM origami, 2 μM TBA15, 2 μM HD22) and Aptarray (~56 nM origami, 2 μM each aptamer) elicited no hemolytic effect after a 3 h incubation, which is similar to the negative control, polyethylene glycol (PEG 8000). We then used thromboelastography (TEG) to evaluate the anticoagulation potential of the

DNA origami-aptamer nanoarray in whole blood samples. The TEG assay measures the global viscoelastic properties of whole blood clot formation under low shear stress (Supplementary Fig. 27), showing the interaction of platelets with the coagulation cascade. In a whole blood sample with buffer added (Control), the lag time (i.e., the interval between kaolin addition and detectable clot formation) was $5.4 \pm 0.1$ min (Fig. 3e, f) and the α angle (the rate of clot formation) was $60.8 \pm 4.0$ degrees (Fig. 3e and Supplementary Fig. 28a). When whole blood from healthy donors was treated with mixtures of aptamers and origami templates (Ori + T + H), we observed a prolonged lag time ($13.9 \pm 0.7$ min) and a reduced α angle ($41.0 \pm 2.0$ degrees). Equimolar concentrations of Aptarray enhanced the lag time to $20.6 \pm 2.8$ min (Fig. 3e, f), with a markedly decreased α angle ($29.4 \pm 4.9$ degrees, Fig. 3e and Supplementary Fig. 28a). The maximum amplitude (MA) values of the TEG tracing obtained following the different anticoagulant strategies were also analyzed (Fig. 3e and Supplementary Fig. 28b). The decreased MA values induced by DNA origami-aptamer nanoarray compared to the control group ($41.87 \pm 4.02$ mm v.s. $62.0 \pm 2.3$ mm) indicates a decline in platelet function and decelerated coagulation. After antidote treatment of the Aptarray-anticoagulated blood, curves and lag times ($5.7 \pm 0.5$ min) similar to the control group were obtained (Fig. 3e, f and Supplementary Fig. 28), indicating a ~100% restoration of thrombin function.

**In vivo anticoagulant activity in mice.** APTT investigation of mice plasma samples (Supplementary Fig. 29) treated by Aptarray or other control groups showed similar trends to the treated human plasma (Fig. 3d and Supplementary Fig. 25a). We next assessed the in vivo thrombin-inhibiting activity of the Aptarray by APTT (Fig. 4a). Mice were treated with buffer, T-16A-H or Aptarray via a single tail vein injection. For neutralization study in vivo, DNA antidotes were administered intravenously to mice followed by Aptarray. Plasma was collected from the treated mice and APTT was measured (Fig. 4a). The Aptarray-treated mouse group exhibited a longer clotting time of $41.7 \pm 2.7$ s post intravenous injection in comparison with buffer treated ($19.4 \pm 0.7$ s) and T-16A-H treated group ($29.7 \pm 1.8$ s). Neutralization by antidote solution through tail injection demonstrated that DNA antidote rapidly and effectively restored the coagulation function (Fig. 4a, antidote-treated: $19.5 \pm 1.0$ s). The in vivo anticoagulation and neutralization were also evaluated by using murine tail-transection bleeding models (Fig. 4b). After the mice received the origami-based anticoagulant through intravenous injection, the tails of the mice were clipped and the blood lost from the tail over the next 15 min was collected and determined. Mice treated with Aptarray (~560 nM, 100 μl) exhibited obvious hemorrhagic effect in response to the trauma (Fig. 4c) compared to the control group and T-16A-H treated group. Administration of DNA antidote in mice completely prevented the excessive bleeding induced by the origami-based anticoagulant and surgical trauma (Fig. 4c).

**Anticoagulation during extracorporeal hemodialysis.** While a potent and antidote-tuned anticoagulant efficacy of the Aptarray was proved in samples of proteins, plasma and whole blood, we sought a solution to the coagulation that can occur during hemodialysis. To this end, we set up a continuous dialysis circulation within an ex vivo circuit, including an artificial kidney (dialyzer) to test the efficacy of the Aptarray (Fig. 5a and Supplementary Fig. 30). Aptarray-containing phosphate-buffered saline (PBS) or blood samples were cycled via a mechanical roller pump through a closed extracorporeal circuit with a dialysis column and reservoir, while dialysate solutions moved through the filter device in the opposite direction and into a waste tank.

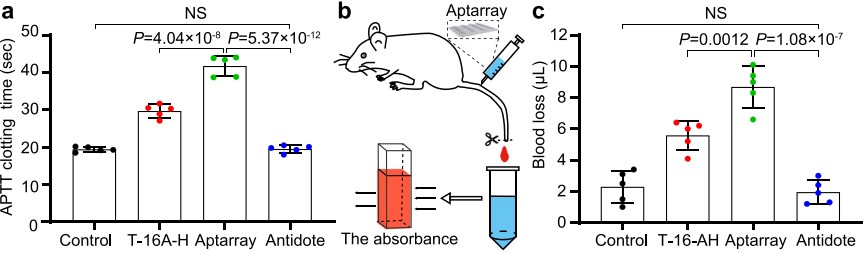

**Fig. 4 Anticoagulation and the neutralization studies in mice. a** Mice ($n = 5$) were treated with buffer (Control), T-16A-H (20 μM, 100 μl) or Aptarray (~560 nM, 100 μl) via a single tail vein injection. Antidotes were administrated intravenously to mice followed by Aptarray for the neutralization study in vivo. Plasma was collected from treated mice and APTT was measured. **b** Schematic drawing of a murine tail-transection bleeding model. **c** Mice ($n = 5$) were treated with buffer (Control), T-16A-H (20 μM, 100 μl) or Aptarray (~560 nM, 100 μl). Antidotes were then administrated intravenously to the anticoagulated animals for neutralization. The tail tips were amputated and the blood loss of mice was measured. Data represent the mean ± s.d. from five (**a**, **c**) independent replicates. Statistical significance (**a**, **c**) was calculated by one-way ANOVA with the Tukey post hoc test. NS, $P > 0.05$. Source data are provided as a Source Data file.

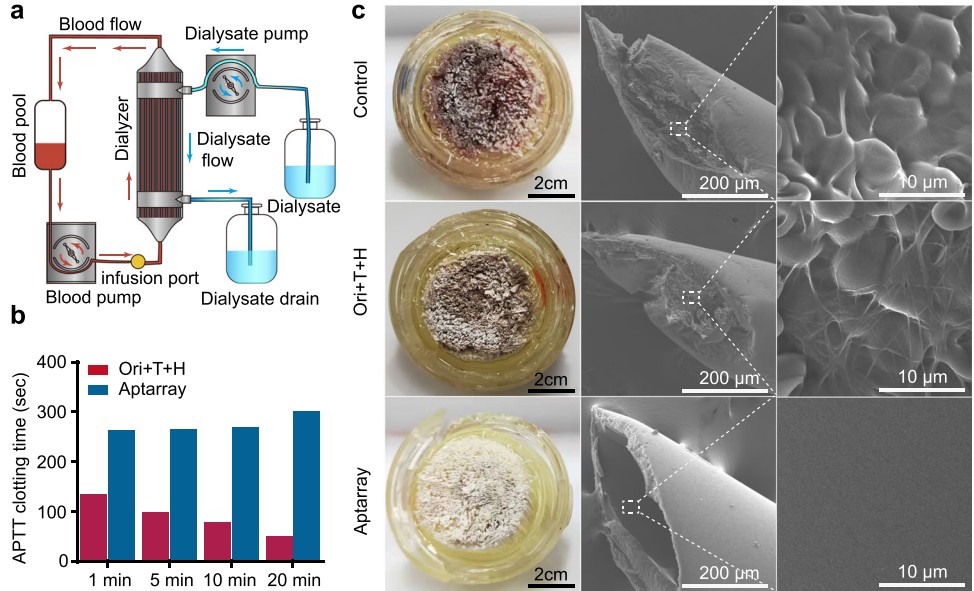

**Fig. 5 Anticoagulant effects during extracorporeal circulation of human blood. a** Schematic drawing of the continuous recirculation within an ex vivo circuit that included a dialysis column and a reservoir (Blood pool). Whole blood from healthy donors is pumped via mechanical roller pump to the dialyzer through the closed extracorporeal circuit, while the dialysate solution flows in the opposite direction through the device and into a waste container (Dialysate drain). The dialyzer and dialysate are heated to ~30 °C. **b** The plasma clotting time (APTT) in the presence of one of the anticoagulant strategies (Ori + T + H or Aptarray) at the indicated time points. The samples were collected from the infusion port. **c** The representative photographs (scale bars, 2 cm) and SEM images (scale bars, 200 μm) of filters collected from the post circulation dialyzer after following the different anticoagulation strategies.

The dialyzer and dialysate solution were heated to ~30 °C (Fig. 5a and Supplementary Fig. 30). In a dialysis circuit, a possible limitation of free aptamers is their high diffusion rate through the dialyzer due to their relatively low molecular weights (<10 kDa). We assessed whether the Aptarray, which possesses increased hydrodynamic sizes and stability, can prolong the circulation time in this extracorporeal circuit (Supplementary Fig. 31). Cy5-labeled free aptamers (TBA15 or HD22) or Aptarrays were injected through an infusion port into the PBS-filled circuit. During 1 h long continuous dialysis cycles, the solutions were collected via the infusion port at several time points (0, 1, 2, 5, 10, 20, 30, and 60 min) and the samples were analyzed by fluorescence spectrometry. Both free aptamers exhibited a rapid clearance from the circuit, with ~15% TBA15 (~4.8 kDa) and ~30% HD22 (~9.4 kDa) remaining after 10 min of circulation. In contrast, the Aptarray were cleared slowly, with ~77% the initial fluorescence retained after the 10 min cycle and ~52% after the 1 h cycle (Supplementary Fig. 31).

We next investigated the anticoagulatory effects of our Aptarray in the ex vivo extracorporeal circuit with fresh human blood from healthy donors (Fig. 5). Blood with only buffer did not show any anticoagulant effect and rapid clot accumulation occurred in the circuit and dialyzer. The APTT assay failed to provide readouts after 1 min circulation due to coagulation (Supplementary Table 1). Scanning electron microscope (SEM) images of filters collected from the post circulation dialyzer showed extensive thrombi comprising by fibrin and blood cells (Fig. 5c and Supplementary Fig. 32). Treatment with the mixture of origami and two aptamers (Ori + T + H) offered modest anticoagulation during the hemodialysis cycling, which is consistent with the aptamer's partial inhibitory effects on thrombin and clot formation and the rapid clearance of the aptamers from the circuit (Fig. 3 and Supplementary Fig. 31). The APTT results obtained from each time point failed to exceed 140 s, and the APTT values decreased with circulation time (Fig. 5b and Supplementary Table 1). A slow clotting process was

observed in the circuit and dialyzer after ~20 min circulation. SEM images of the post circulation dialyzers revealed fibrin and blood cell accumulation in the tiny hollow filters (Fig. 5c and Supplementary Fig. 32). In contrast to the origami plus two aptamers groups, blood samples treated with Aptarray displayed significantly higher (>250 s) and more stable APTT values (Fig. 5b and Supplementary Table 1), with no observable clot appearing during the 1 h circulation. A slight increase of APTT values is due to the dilution of the whole blood in the extracorporeal circuit by dialysate solutions (Supplementary Fig. 33). The presence of Aptarray effectively prevented clot formation in the dialyzer, and no obvious fibrinous or cellular debris was observed in the hollow filters from the dialyzer after treatment (Fig. 5c and Supplementary Fig. 32).

**Safety assessment of the Aptarray.** Therapeutic agents in clinic use that prevent blood clotting in hemodialysis often engender post-treatment bleeding[1,4]. Our neutralization studies in fibrinogen-containing solutions (Supplementary Fig. 24), human plasma (Fig. 3d), whole blood samples (Fig. 3e, f) and a mouse model (Fig. 4) demonstrate that the complementary DNA strand antidote can rapidly and effectively neutralize the anticoagulant activity of Aptarray, reducing the risk of hemorrhage.

To assess biocompatibility in vitro, we compared the viabilities of human embryonic kidney HEK293T cells and mouse brain endothelioma b.End3 cells in the presence of Aptarray (0–56 nM) for 24 h or 48 h (Supplementary Fig. 34). No cytotoxicity was observed in either cell type after treatment. Furthermore, we tested serum indices of liver and kidney function in mice and found that the parameters, including alanine transaminase (ALT), aspartate transaminase (AST), creatinine (Cr) and blood urea nitrogen (BUN), were in the normal ranges after intravenous injection of the Aptarray (Supplementary Fig. 35). To test the immunogenicity of our Aptarray in vivo, we measured the blood levels of interleukin-6 (IL-6) and tumor necrosis factor-α (TNF-α) after intravenous injection of buffer (Control), the mixture of the two aptamers and origami (Ori + H + T), or Aptarray into healthy mice (Supplementary Fig. 36). The serum IL-6 and TNF-α were both within the normal ranges, indicating that the Aptarray did not trigger an immune response at the tested dose. Histological examination of the major organs (heart, liver, spleen, lung and kidney) revealed no observable toxicity 24 h post-injection of the mixture of two aptamers (H + T), the mixture of origami and aptamers (Ori + H + T) or Aptarray (Supplementary Fig. 37).

Hemodialysis is a standard treatment for patients with renal failure. A recent study demonstrated that DNA origami nanostructures can protect against the nephrotoxicity of reactive agents in a kidney injury animal model[34]. We found that the Aptarray exhibited reactive oxygen species (ROS)-eliminating effects when incubated with ABTS molecules, an in vitro indicator of antioxidant capacity (Supplementary Fig. 38a, b). We next used HEK293T cells to investigate the effects of the Aptarray on oxidative stress induced by the addition of $H_2O_2$. At $H_2O_2$ concentration of 250 μM and 500 μM, the viability of cells in the Aptarray-treated groups was significantly higher than in cells treated with $H_2O_2$ alone (Supplementary Fig. 38c), corroborating that the DNA nanostructures relieve oxidative stress at the cellular level.

## Discussion

In this work, we describe a strategy that uses a DNA origami-based aptamer nanoarray as an anticoagulant for hemodialysis treatment. Our origami-based approach allowed the construction of a structurally well-defined platform with an addressable surface. This platform enabled the organization of two types of thrombin-binding DNA aptamers to form a pre-designed nanoarray and elicit synergistic effects of recognition and inhibition of thrombin. The aptamer-loaded DNA origami displays potent anticoagulation in human plasma, fresh whole blood and mice in vivo. In contrast to heparin-like compounds, the most potent and frequently administered intravenous anticoagulants, our Aptarray generates a robust but controlled anticoagulation, which can be reversed by sequence complementary oligonucleotides of the aptamers. The increased hydrodynamic sizes by the origami template can prolong the circulation time of Aptarray, which induce more efficient anticoagulatory effects compared with free bi-aptamers tethered with polyA linkers in vivo. In an ex vivo extracorporeal circuit model with fresh human blood, the Aptarray effectively inhibited blood clot formation, underscoring the potential of this anticoagulation strategy for future in vivo hemodialysis. Compared to free aptamers with small molecular weights, the Aptarrays were retained more effectively in the extracorporeal circuit to sustain a long-lasting anticoagulation effect in hemodialysis model. Additionally, the DNA origami nanostructures can serve as ROS scavengers in vitro, potentially alleviating damage in acute kidney injury mouse[34]. The current DNA origami-based anticoagulant could be further advanced by integrating multiple aptamers that inhibit procoagulant proteases in coagulation pathway for multi-step prevention of blood clotting. Other clinical procedures, such as hemofiltration, hemodiafiltration, hemoperfusion and cardiopulmonary bypass, require filtration and/or extracorporeal oxygenation of the blood, which also pose the risk of activating the coagulation cascade. Our potent, antidote-controllable, DNA origami-based nanoscale anticoagulants are applicable in those extracorporeal procedures. With the advances in mass production strategies of DNA nanostructures[35], the cost of origami-based anticoagulant can be greatly reduced, allowing for future clinical studies. This DNA origami platform will open an avenue for producing a safe and effective anticoagulant for the extracorporeal treatment of renal failures and other diseases.

## Methods

**Materials.** All oligonucleotides including staple strands, capture strands and aptamers were purchased from Integrated Invitrogen (Shanghai, China). The origami staple strands were stored in tubes with concentrations normalized to 100 μM, and were used without further purification. The concentration of each strand was estimated by measuring the UV absorbance at 260 nm. Thrombin from human plasma (T6884) was purchased from Sigma-Aldrich. Dialyzers (AEF-03) were purchased from AsahiKASEI. Unfractionated heparin (UFH, Changzhou Qianhong Bio-pharma Co. Ltd.) and low molecular weight heparin (LMWH, QILU Pharmaceutical Co. Ltd.) were obtained from the clinical laboratory of the second affiliated hospital of Xi'an Jiaotong University.

**Self-assembly of the DNA origami templates.** Production of M13 bacteriophage single-stranded DNA was according to Douglas et al.'s methods[36]. Briefly, JM109 E. coli were cultured in 2 × YT medium (5 mM MgCl_2) and placed in a shaker at 37 °C. When the optical density (OD 600) reached 0.5, M13 phages (p7249) were mixed with the bacteria, and then cultured for 5 h. The culture was collected and centrifuged at 6000 × g for 30 min to remove the bacteria pellets. NaCl (30 g/l) and PEG (40 g/l) were added to the supernatant (containing phages), and the mixture was incubated on ice for 1 h. The phage pellet was collected after 30 min centrifugation at 10,000 × g, and suspended in Tris-HCl (10 mM pH 8.5). The phage solution was incubated with NaOH (0.2 M) and SDS (1%) at 25 °C for 3 min. After the addition of potassium acetate (3 M, pH 5.5), the solution was incubated on ice for 10 min and centrifuged (12,000 × g, 30 min). The ssDNA (7249 nt) containing supernatant was collected and precipitated in ethanol (70%) on ice for 2 h. After centrifuging at 12,000 × g for 30 min, the DNA pellet was collected and washed in ethanol (70%) and then resuspended in Tris-HCl (10 mM, pH 8.5). The concentrations of ssDNA were determined by UV–Vis spectrometry (UV-2450, Shimadzu, UVProbe 2.61 software).

Rectangular shaped DNA origami structures were assembled according to Rothemund's methods[19] with several modifications. The positions and sequences of the staple/functional strands are shown in different colors in Supplementary Figs. (1–7, 14) and Supplementary Table 2. A molar ratio of 1:5:5 among the long

viral ssDNA M13mp18 (20 nM), the short staple strands (100 nM) and the capture strands (100 nM) was used. DNA origami was annealed and assembled in TAE-$Mg^{2+}$ buffer (Tris, 40 mM; Acetic acid, 20 mM; EDTA, 2 mM; and Magnesium acetate, 12.5 mM; pH 8.0) in an Eppendorf thermocycler (Eppendorf China) by slowly cooling from 95 °C to 25 °C at a rate of 10 min/°C. The resulting rectangular DNA origami structures were separated from excess staple strands using Amicon Ultra-0.5 ml 100 kD centrifugal filters (Millipore). The purified origami solution was collected and characterized using 1% agarose gel electrophoresis and atomic force microscopy (AFM).

**Preparation of anticoagulant assemblies**. Purified DNA origami was mixed with thrombin-binding aptamers (TBA15 and HD22) with a ratio of three aptamer molecules for one docking site. The mixture was annealed in 1 × TAE-$Mg^{2+}$ buffer containing 100 mM NaCl from 45 °C to 25 °C at a rate of 5 min/°C for six cycles.

Purified and concentrated origami-aptamers assemblies were achieved by performing poly (ethylene glycol)-induced depletion[30]. Briefly, origami-aptamers samples were mixed with PEG 8000 (15%), NaCl (500 mM) and $MgCl_2$ (10 mM), then the mixture was centrifuged at 25 °C for 40 min at 16,000 × $g$. Removing supernatant, the pellet was dried by a vacuum centrifuge concentrator (CV200, Beijing Jiaimu, China) and resuspended in 1× TAE-$Mg^{2+}$ buffer. The purified nanoarray solution was collected and quantified by UV–VIS spectrophotometer. The nanoarray was then characterized using 1% agarose gel electrophoresis and AFM.

**AFM characterization**. For DNA origami structures, 10 μl of sample was deposited onto a freshly cleaved mica and left to absorb to the surface for 10 min. The sample was subsequently added with 40 μl TAE-$Mg^{2+}$ buffer and then covered by the liquid cell for imaging. AFM imaging of DNA nanostructures before and after aptamer loading was performed in ScanAsyst mode (Mutimode-8, Bruker). The images were collected and processed using Bruker NanoScope Analysis 1.9 software.

**Gel electrophoresis**. DNA origami-aptamer assemblies were separated on a 1% agarose gel (EtBr stained, running buffer 0.5× TBE with 11 mM $Mg^{2+}$, 15 V cm$^{-1}$) and imaged under UV irradiation.

**Hydrodynamic measurements**. The hydrodynamic radius distribution and zeta potential of bare origami and DNA origami-aptamer nanoarray were examined by dynamic light scattering instrument (DLS, Malvern) and dispersion technology software (Zetasizer Nano ZS 7.11) with a standard setting.

**Monitoring of the clotting reactions**. The clotting reaction was monitored by the changes of scattered light intensity from the sample cuvette with a fluorescence spectrometer (Cary Eclipse, Agilent Technologies). OriginPro 9.1 software was used to collect data. Briefly, nanoarrays were diluted by clotting reaction buffer (20 mM Tris, 150 mM NaCl, 5 mM KCl, 1 mM $MgCl_2$, 1 mM $CaCl_2$, pH 7.4) to the indicated concentrations. One-hundred seventy-eight microliters of Aptarrays were incubated with thrombin (12 nM) at 25 °C for 5 min. Aptamers or the mixtures of aptamers and origami were also treated with thrombin as control groups. Fibrinogen (10 mg/ml, 20 μl) was quickly added to these reaction mixtures and the scattering intensities were measured at 650 nm for ~9000 s on a fluorescence spectrophotometer. All the clotting times were normalized based on the standards.

**Blood**. Human plasma samples were obtained from Peking University Fourth School of Clinical Medicine under the protocol (201904-06) approved by Beijing Jishuitan Hospital Institutional Review Board. For thromboelastography (TEG) and extracorporeal hemodialysis circuit, blood was drawn from healthy volunteers (informed consent was obtained from all subjects). The study was approved by the Institutional Ethics Committee, National Center for Nanoscience and Technology of China (NCNSTIEC0068-0109). Blood draw procedures were achieved in accordance with institutional guidelines. Blood was anticoagulated with 3.2% sodium citrate unless otherwise noted.

**Coagulation assay**. Fresh blood from healthy volunteers was collected and plasma was prepared by centrifuging citrated human blood at 2000 × $g$ for 15 min at 4 °C to remove blood cells. The clotting reaction of human plasma was monitored by an automatic coagulation analyzer. Thirty microliters Aptarray (with or without antidote treatment) were mixed with 270 μl plasma at 25 °C for 5 min. Buffer, TBA15, HD22, Ori + T + H, heparin and LMWH solutions, were also incubated with plasma as control groups. Activated partial thromboplastin time (APTT) assays and prothrombin time (PT) were performed using a Sysmex CS-5100 System analyzer (Siemens, Germany) according to the manufacturer's instructions.

**Thromboelastography (TEG) assay**. Firstly, citrate anticoagulated whole blood (320 μl) from healthy donors was mixed with 35 μl samples and kaolin (10 μl) to initiate the clotting process. The four different treatments were TAE-$Mg^{2+}$ buffer, the mixture of aptamers and origami (Ori + H + T, aptamers 20 μM, origami

~560 nM), Aptarray (~560 nM) and antidote-neutralized Aptarrays (fivefold antidotes pre-incubated Aptarray for 5 min). After the addition of $CaCl_2$ (20 μl, 0.2 M), the mixture was immediately transferred to a TEG cup, and the assay was run at 37 °C according to the manufacturer's instructions. Clot formation was measured with a Thromboelastograph Analyzer (Haemonetics) until a stable clot was formed (~60 min). The lag time, α angle and the maximum amplitude were automatically calculated by TEG analytical software 4.2.3 (Haemonetics).

**Extracorporeal hemodialysis circulation**. Briefly, the circuit consisted of a custom-designed reservoir, two mechanical roller pumps (Baoding Shenchen peristaltic pump company, China) and a 30-ml dializer for hemodialysis (AEF-03), which were all connected via homemade tubing (TianJing Hanaco medical company, China).

To test the clearance, Cy5-labeled aptamers (TBA15 or HD22, 1 μM) or Aptarrays (origami ~28 nM, containing 1 μM TBA15 and 1 μM HD22) were injected through an infusion port into PBS solutions that filled in the circuit. After 1 h period of continuous dialysis circulations, solutions were collected via the infusion port at indicated time points (0, 1, 2, 5, 10, 20, 30, 60 min). The samples were analyzed for fluorescence intensity at 650 nm on a fluorescence spectrophotometer (Agilent Technologies).

Blood was pumped via a roller pump to dializer through the closed extracorporeal circuit; while dialysate (TRHD company, TR-2, Liquid A: $Na^+$ 138 mM, $K^+$ 2.0 mM, $Ca^{2+}$ 1.5 mM, $Mg^{2+}$ 0.5 mM, $Cl^-$ 109.5 mM, $HCO_3^-$ 32 mM, $CH_3OO^-$ 3.0 mM; Liquid B: $Na^+$ 35 mM, $HCO_3^-$ 35 mM) flowed through the device and disposed. To provide temperature control, a constant temperature water bath was used to contain the dializer, and a thermostatic magnetic stirrer was used to hold a flask that contained the dialysate solution. The temperature of the circulating blood was maintained at ~30 °C. Blood flow rate was 10 ml/min, while dialysate flow rate was 50 ml/min. Prior to the addition of blood and dialysate, the circuit was primed with PBS and circulated continuously at a flow rate of 10 ml/min for ~30 min.

Fifty milliliters of whole blood was collected from individual healthy donors and anticoagulated with 3.2% sodium citrate. After draining the PBS from the circuit, equivalent volume buffer, the mixture of aptamers and origami (Ori + H + T) or Aptarray-treated blood was added. $CaCl_2$ (1 M, 322.5 μl) was injected into the circuit through the infusion port and mixed with the blood for 5 min to neutralize sodium citrate. The circulation was initiated (time = 0 min) at a flow rate of 10 ml/min with dialysate flowed through the filter device in an opposite direction at 50 ml/min. Blood samples (~2 ml) were withdrawn from the circuit at 1, 5, 10, 20 min after circulation was initiated for APTT measurement. After 60 min of circulation, circuit blood was removed and the dializer was then rinsed with PBS and fixed overnight with 2.5% glutaraldehyde.

Prefixed dializer was mechanical crashing and the hollow filters were separated. These fibers were cut into pieces and dehydrated by gradient ethanol (10%, 30%, 60%, 90%, 100%). The fibers were subsequently submerged in liquid nitrogen for 5 min, and then were affixed to circular metal stubs. The samples were obtained Pt-sputtering in an ion sputter coater for 20 s. The micrographs were obtained with scanning electron microscope (SEM, Hitachi SU8200, Japan). The data were collected and processed using Hitachi SU8200 ver.1.18.

**Cell culture**. All cell lines were purchased from the American Type Culture Collection (Manassas, VA, USA) unless stated otherwise. Human embryonic kidney 293T cells (HEK293T), mouse brain endothelial cells (bEnd.3) were maintained in high-glucose DMEM supplemented with 10% FBS, 100 U/ml penicillin and 100 U/ml streptomycin. Cell line authentication was performed by short tandem repeat DNA profiling and comparison with a reference database. The cells were cultured at 37 °C, 5% $CO_2$ and were routinely tested for mycoplasma contamination.

**Cell viability assay**. The cytotoxicity of the origami-aptamer assemblies was assessed with a cell-counting kit (cck-8, Dojindo, Japan) containing a highly water-soluble tetrazolium salt (WST-8) [2-(2-methoxy-4-nitrophenyl)-3-(4-nitrophenyl)-5-(2, 4-disulfophenyl)-2H-tetrazolium, monosodium salt]. After seeding 96-well plates and culturing overnight, the HEK293T and bEnd.3 cells were incubated with Aptarray (final concentrations ranging from 0 to 56 nM and diluted with culture media) for 24 h before being washed with PBS. Next, the cells were incubated with fresh serum-free medium containing 0.5 mg/ml WST-1 for 1 h at 37 °C for the cytotoxicity assay. Absorbance values at 450 nm were measured using a microplate reader (TECAN, infinite M200, Switzerland).

**Scavenging ABTS radicals with Aptarray**. The free radical scavenging capability of Aptarray was tested based on the reduction of •ABTS$^+$ radicals using the ABTS radical cation decolorization assay according to previous report[34]. The UV–Vis spectra of the pure solution of •ABTS$^+$ radicals and •ABTS$^+$ radical solution with Aptarray were determined. The inhibition rate of •ABTS$^+$ radicals was calculated based on the ratio of neutralized •ABTS$^+$ radicals to overall radicals using the absorbance at 734 nm. All measurements were carried out in five independent experiments.

**Protection of cells exposed to $H_2O_2$ with Aptarray**. HEK293T cells were seeded into a 96-well plate at $10^6$ cells per well and then incubated for 24 h at 37 °C under 5% $CO_2$. Aptarray (final concentration in each well, 9.3–56 nM) was added to the cells and they were incubated for 30 min. The cells were then incubated with $H_2O_2$ (final concentration in each well, 0–2000 μM) for 24 h at 37 °C with 5% $CO_2$. Cytotoxicity assay by cell-counting kit was then performed to evaluate cell viability.

**Animal study**. All animal studies were performed in accordance with ARRIVE guidelines, with the approval of the Institutional Animal Care and Use Committee of National Center for Nanoscience and Technology.

Six to eight-week-old BABL/c mice were obtained from Vital River Laboratory Animal Technology Co. Ltd (Beijing, China) and housed with a 12 h light/dark cycle at 22 °C (40% relative humidity), and with food and water ad libitum.

**Hemolytic analysis**. Fresh blood samples from BABL/c mice were anticoagulated with 3.2% sodium citrate and immediately centrifuged (3000 × g, 10 min, and 4 °C) to remove serum. The red blood cells (RBCs) collection procedures were performed in compliance with the relevant laws and institutional guidelines. The RBCs were diluted with sterile isotonic physiological buffer to obtain an RBC stock suspension. Aptamers, the mixture of DNA origami and aptamers, Aptarray, PEG 8000 (negative control) or triton X100 (positive control) was added to the RBC suspension in 1.5 ml tubes, respectively. After 3 h of incubation at 37 °C, each of the mixtures was centrifuged (1000 × g for 10 min). Hemolysis activity was determined by measuring hemoglobin absorption at 560 nm ($A_{560}$) in the supernatant.

**APTT assay in mice plasma and in vivo**. For APTT investigation in plasma, fresh blood from healthy BABL/c mice was collected. Mice plasma was prepared by centrifuging citrated blood at 3000 × g for 10 min at 4 °C to remove blood cells. Fifty microliters of plasma was mixed with 50 μl TriniClot APTT reagent for 5 min incubation, followed by 16.7 μl Aptarray (~560 nM, containing 20 μM TBA15 and 20 μM HD22), T-16A-H (20 μM) or other control groups at 37 °C for 5 min. After the addition of 50 μl $CaCl_2$ to initiate the coagulation, the clotting reaction of mice plasma was monitored by a semi-automatic coagulation analyzer (SC 40, STEELLEX).

For in vivo anticoagulation, BABL/c mice were administrated of buffer (100 μl), the mixture of origami and aptamers (Ori + T + H, aptamers 20 μM and origami ~560 nM), T-16A-H (20 μM) or Aptarray (~560 nM, containing 20 μM TBA15 and 20 μM HD22) via a single tail vein injection. Blood samples collected from mice were anticoagulated with 3.2% sodium citrate at 5 min. Whole mice blood was centrifuged at 3000 × g for 10 min at 4 °C to remove blood cells. The collected plasma sample was mixed with 50 μl TriniClot APTT regent for 5 min incubation. After the addition of 50 μl $CaCl_2$ to the solution, the clotting reaction of mice plasma was monitored by a semi-automatic coagulation analyzer.

**Tail transection and bleeding studies in mice**. The mice received tail-vein injections of buffer (100 μl), T-16A-H (20 μM, 100 μl) or Aptarray (~560 nM, 100 μl). Five minutes later, the mice were anesthetized by intraperitoneal injection of phenobarbital (100 mg/kg) and the distal 1 mm of the tails were amputated. The tail was immersed for 15 min in 1 ml of saline warmed to 37 °C. Blood loss was determined as previously described[37], by measuring the absorbance of saline at 560 nm and comparing the result to a standard curve constructed from known volumes of mouse blood. For neutralization study, the mice first received Aptarray (~560 nM, 100 μl) by tail-vain injection. Fivefold DNA antidote solution was then administered by intravenous injection. Blood loss from the tail wound was collected for 15 min after tail transection, and was measured as described previously.

**Safety assessment**. Mice were received intravenous administration of buffer, aptamer mixtures (20 μM, 100 μl), the mixture of origami and aptamers (Ori + H + T, aptamers 20 μM, origami ~560 nM, 100 μl) and Aptarray (~560 nM, 100 μl). Blood samples were collected at 24 h post-injection and sent to the laboratory of Pony Testing International Group (Beijing, China) for analysis of the levels of alanine transaminase (ALT), aspartate transaminase (AST), creatinine (Cr), and blood urea nitrogen (BUN). The plasma levels of TNF-α and IL-6 were analyzed by corresponding ELISA kits (Cloud-Clone Crop). Major organs (including the heart, liver, spleen, lung, and kidney) were harvested from mice for hematoxylin and eosin staining to evaluate morphological changes caused by the administration of DNA materials.

**Statistical analysis**. The methods for statistical analysis and sizes of the samples (n) are specified in the results section or figure legends for all of the quantitative data. All values are presented as mean ± s.d. with the indicated sample size. No samples were excluded from analysis. Statistical differences were determined using one-way analysis of variance (ANOVA) followed by the Tukey post hoc test for three or more groups. Statistics were performed using GraphPad Prism 8 and SPSS 24.

**Reporting summary**. Further information on research design is available in the Nature Research Reporting Summary linked to this article.

## Data availability

The data that support the findings of this study are available within the paper and its Supplementary Information files. Additional data and files are available from the corresponding author upon reasonable request. Source data are provided with this paper.

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

## Acknowledgements

This work is supported by the National Natural Science Foundation of China (22025201, 32071389, 31700871, 21708004, 21773044, and 51761145044), Beijing Municipal Science & Technology Commission (Z191100004819008), the National Basic Research Program of China (2016YFA0201601, 2018YFA0208900), Key Research Program of Frontier Sciences, CAS, Grant QYZDBSSW-SLH029, the Strategic Priority Research Program of Chinese Academy of Sciences (Grant No. XDB36000000), CAS Interdisciplinary Innovation Team and K.C. Wong Education Foundation (GJTD-2018-03).

## Author contributions

B.D. and Q.J. conceived and designed the experiments. S.Z., Q.J., R.T., J.W., S.L., Y.W., Y.S, Z.W., Y.L., M.W., Y.Z., and T.W. performed the experiments. S.Z., Q.J., R.T., and B.D. collected and analyzed the data. J.W., Q.L., Y.Z., H.Z., Y.S., H.C., Y.G., and J.S. provided suggestions and technical support on the project. B.D. supervised the project. Q.J., S.Z., and B.D. wrote the manuscript. All authors discussed the results and commented on the manuscript.

## Competing interests

The authors declare no competing interests.
