## [Peer Review File · Nature Communications]

Reviewers' comments:

Reviewer #1 (Remarks to the Author):

I found the manuscript to be well written and interesting because brings DNA nanotechnology to an new application area.

The manuscript makes a significant step towards assessing the potential of DNA nanostructure as an anticoagulation treatment but it is important to note that the paper builds on published results that characterised the increase in thrombin affinity possible by localising two types of aptamer at an optimum spacing. I'm not sure that I would go as far as to say that the paper will influence thinking in the field but as a well executed exploration of a new application area I would like to see it published.

Broadly the experiments support the conclusions (although see point 1a).

Major comments:

1. A key message of the manuscript is that control over type, arrangement and density of thrombin aptamers allows potent inhibition of clotting.

a) the manuscript doesn't include data on density (or number) of aptamers.

The manuscript shows that two types of aptamer localised on an origami tile is a more effective anticoagulant than free aptamers at the same concentration, the manuscript also shows that aptamers on an origami tile are cleared more slowly than free aptamers (a useful property) but it doesn't distinguish between colocalisation and density. If two aptamers were tethered together by a simple double-stranded linker (free from all of the complication of a DNA origami) - what effect would they have on coagulation? Is the effect of multivalency important or not? This is a useful control experiment and even if a simple aptamer pair is an effective reagent it is unlikely to match the reduced clearance rate of the DNA origami tile.

b) the manuscript doesn't put the results in context of what has already been done with origami and thrombin aptamers nor does it make comparisons with other strategies (it is hard to assess how useful this approach will be)

work on type and arrangement of aptamers is largely consistent with published results.

of I would like to see a better survey of the work so far on thrombin decorated DNA / RNA nanostructures and a comparison with results from other approaches, for example using decorated nanoparticles.

Krissanaprasit et al (ref 24, <https://doi.org/10.1002/adma.201808262>) is cited as an example of 'nanostructures with rationally designed geometric features, precise spatial addressability and remarkable biocompatibility' but not as an example of a device designed to inhibit coagulation. Chhabra et al is not cited (<https://doi.org/10.1021/ja072410u>).

Chhabra et al (<https://doi.org/10.1021/ja072410u>) show attachment of thrombin to DNA origami tiles using a row of identical aptamers (given this result it is perhaps surprising that the control tiles in the manuscript don't show any thrombin binding e.g. in the AFM assay shown in Fig S6,7, 8). Can the authors account for this difference? What was the thrombin concentration used in AFM assays.

Rinker (ref 21) looked at the effect of spacing between thrombin aptamers on binding affinity and found a 5.4 nm spacing had a 10-fold higher affinity for thrombin than a 24 nm spacing. This result should be noted because the results in the manuscript on aptamer spacing confirm the results of Rinker.

Similarly, the use of the reverse complement to release thrombin has been described elsewhere - this is a useful property but not a surprising one, the precedent in the literature should be described.

Minor comments:

2. Too many similar ways to describe the same thing leads to confusion (DNA origami nanoarray for anticoagulation, aptarray, DNA nanoanticoagulants, DNA-origami based bi-aptamer arrays). Aptamer on its own called TBA15 / HD22 but, when mixed with origami called T/H, when both aptamers bound aptarray but when only one bound and the other present called TBA15-loaded + HD22 or HD22-loaded + TBA15. Please find a simpler nomenclature so that it is easier to interpret control

experiments. Also note that the caption to Fig S18 says aptamer loaded and the figure says aptamer-loading.

3. Fig. 2.

Fig. 2a (and Fig S13). What temperature was the gel run at? The Alexa 488 channel shows a fast running smear at the bottom of the gel, I would prefer to see the whole gel because I wonder if the smear is evidence of the aptamer dissociating from the origami as the gel is run.

Fig. 2bc. Why not show both aptarray and origami result in this figure (i.e. move traces from Fig. S15)

4. Fig. 3bc (and Fig. S17-21 especially Fig. S17). Thrombin activity is described as the initial rate or the linear part of the trace. This requires further explanation. In Fig. S17 for example, I can't see any linear portion of the trace. I think it is worth a more careful consideration of how to interpret these results.

5. In Fig. 3bc, control experiments were done with 20 nM aptamer. Aptarray experiments were done at 'equivalent aptamer amounts' - it says that this is 0.6 nM in the caption but it would be useful to add this to the text. It should also be made clear what the dose-dependent concentrations are in Fig. S19 (I assume that these are aptamer concentrations not aptarray concentrations).

Reviewer #2 (Remarks to the Author):

This article reports the design and characterization of a DNA-origami platform with anticoagulant activity. Using a DNA origami as a scaffold, the authors were able to spatially introduce arrays of thrombin-binding aptamers positioned at either 68, 46, 24 or 5.4 nm from each other. They found that the 5.4 nm array (distance between two thrombin aptamers TBA and HD22) provides a better binding towards thrombin and therefore displayed a stronger inhibition of thrombin's catalytic activity, an enzyme responsible of coagulation. Using this nanosystem, they demonstrated its anticoagulation properties in human plasma, in whole blood and with mice in vivo study. They further investigated the capacity of using this new nanosystem for extracorporeal hemodialysis while providing technical and safety assessments.

This paper reports an interesting work on a clinically relevant application. Few papers on DNA origami display such a nice clinical validation. The experiments are well done and support the conclusion and the results and figures are very nice.

***My main concern with this paper is that the enhanced effect obtained from this complex origami strategy, can typically be obtain via a much simpler chemical strategy: by chemically attaching together two inhibitors that binds the same target (see e.g. the classic SAR by NMR paper: <https://science.sciencemag.org/content/274/5292/1531>).

This strategy is much simpler and has also already been successfully realized for the TBA-HD22 system employed by the authors: you can find this info on wikipedia:

“Avidity effect of TBA and HD22:

Similar to antibody, aptamers TBA and HD22 show avidity effect against thrombin after dimerization. When TBA and HD22 are conjugated with an optimal linker[19][20] or co-printed on the sensor surface with an optimal density,[21] the affinity against thrombin could be significantly enhanced by 100 to 10,000 fold. Furthermore, the dimerization improves the anticoagulant activity as well. The TBA-HD22 construct (linked with 16-mer polyA) shows significant improvement both in the assay of activated partial thromboplastin time, clotting time and thrombin-induced platelet-aggregation.”

Unfortunately, the authors fail to mention this fact and did not provide an hypothesis to explain why their origami strategy is working...

If the authors could demonstrate that their origami platform (with its ability to attach aptamers at specific distance from each other) performs better or as good as a simple conjugation strategy (e.g. using the optimal TBA-HD22 conjugate), I may recommend the publication of this article. But if this is not the case, I believe that this strategy remains way too complex, expensive (complex synthesis, assembly and purification) and cumbersome to be useful/commercialized.

Of note, as mentioned by the authors, the origami remains for a longer period in blood due to its bigger size. This effect could also be easily obtained by attaching the small TBA-HD22 conjugate to albumin via the presence of an hydrophobic moiety.

Here are other suggestions that I think may improve this paper:

1) Multiple typos are found throughout the manuscript. This includes missing comma, inverted letters (ex. aptamers vs aptamers), missing words (ex. “which is consistent with the aptamer’s partial inhibitory effects”), double wording (ex. “In contrast to from traditional nanoparticles”, missing letter. A closer look at typos should be done prior to submission.

2) On page 5 and 6, authors mentioned briefly why the DNA origami with 5.4 nm between each aptamer provides a more efficient binding with thrombin compared to DNA origami with distances of 64, 46 and 24 nm. They mentioned a potent synergistic effect. Authors should develop more on the chemical nature of that proposed synergistic effect and mentioned information about the distance between the two exosites and how this can enable both aptamers to bind simultaneously the thrombin therefore increasing the affinity of thrombin for the Aptarray. As mentioned above, please cite the literature about the “Avidity effect” (how to obtain a “better” inhibitor with smaller KD) by attaching two recognition elements to each other...

3) On page 6, authors measured the size and the zeta potentials of their DNA origamis and provided some values. The authors should explain why they measured the zeta potential and the importance of this measure to characterise their two assemblies.

4) On page 7, authors mentioned two specific measurements to quantify anticoagulation properties. These two measurements are called APTT (Activated Partial Thromboplastin Time) and PT (Prothombin Time). A better explanation of these two values should be done in order to better appreciate the results. Authors should explain these two parameters the same way they explained the concept of thromboelastography (TEG) on page 8 (Fig. S24).

5) On figure S12, we don’t see clearly the band of the redundant aptamers strand that are eliminated by PEG-induced precipitation. A better light contrast should resolve that problem. On figure S17, the x axis is missing. On the DNA sequence at the end of the SI, authors must mention the 3’ and 5’ extremities.

6) Throughout the manuscript, authors use different names to design the Aptarray. Sometimes it is referred as DNA nanoanticoagulant, DNA origami-aptamer nanoarray, origami-based anticoagulant, DNA origami-based aptamer nanoarray, DNA origami-based anticoagulants, DNA origami-based nanoagent. Several terms for the same thing only create confusion therefore one term should be prioritized to describe the Aptarray.

Best regards

POINT-BY-POINT RESPONSE TO THE REVIEWERS' COMMENTS

We thank the reviewers for their constructive and valuable comments. Below, the reviewers' comments are shown in **bold**, and our responses are in the standard typeface. Sections highlighted in **yellow** represent wording that has been altered/inserted into the revised manuscript. The corresponding sections in the revised manuscript are also highlighted in **yellow**.

In addition to these changes, a number of changes to the wording and grammar have been made to improve the readability of the manuscript. These have not all been highlighted, to avoid unnecessary clutter in the revision, but any substantive changes have been marked. Other than the Figures in the revised main text and supplementary file, additional figures presented in this response letter are labelled as **Figure Rs**.

Reviewers' comments:

Reviewer #1 (Remarks to the Author):

I found the manuscript to be well written and interesting because brings DNA nanotechnology to an new application area.

The manuscript makes a significant step towards assessing the potential of DNA nanostructure as an anticoagulation treatment but it is important to note that the paper builds on published results that characterised the increase in thrombin affinity possible by localising two types of aptamer at an optimum spacing. I'm not sure that I would go as far as to say that the paper will influence thinking in the field but as a well executed exploration of a new application area I would like to see it published.

Broadly the experiments support the conclusions (although see point 1a).

Major comments:

1. A key message of the manuscript is that control over type, arrangement and density of thrombin aptamers allows potent inhibition of clotting.

a) the manuscript doesn't include data on density (or number) of aptamers.

The manuscript shows that two types of aptamer localised on an origami tile is a more effective anticoagulant than free aptamers at the same concentration, the manuscript also shows that aptamers on an origami tile are cleared more slowly than free aptamers (a useful property) but it doesn't distinguish between colocalisation and density. If two aptamers were tethered together by a simple double-stranded linker (free from all of the complication of a DNA origami) - what effect would they have on coagulation? Is the effect of multivalency important or not? This is a useful control experiment and even if a simple aptamer pair is an effective reagent it is unlikely to match the reduced clearance rate of the DNA origami tile.

Our response 1a):

We would like to express our gratitude to the Reviewer for this excellent suggestion for additional controls. The simple aptamer pair controls considerably strengthen our hypothesis that two aptamers colocalized on origami elicited a potent synergistic effect on thrombin recognition and inhibition.

To comply with the Reviewer's insightful suggestion, we have now repeated the experiments of clotting reactions and added two additional control groups: aptamers were tethered with 5-mer or 16-mer polyA linkers (*ChemBioChem*, 2007, 8, 2223; *Journal of Thrombosis and Haemostasis*, 2008, 6, 2105).

Through light scattering measurements (**revised Figure 3b-c**), aptamers tethered with 5-mer or 16-mer polyA linkers (T-5A-H, T-16A-H) resulted in much slower fibrin formation velocities, in comparison with the mixtures of the aptamers and rectangular origami templates at equivalent aptamer amounts (Ori + T + H). Thrombin inhibitory performance can be significantly increased ($98.3 \pm 0.2\%$) by arraying aptamers in a multivalent form on the DNA origami template with bi-aptamer distances of 5.4 nm. The results demonstrated that the effect of colocalization induced-multivalency of aptamers is important for thrombin inhibition. It also indicated that the precisely organized aptamer array on the DNA origami elicited a potent synergistic effect on thrombin recognition and inhibition.

Revised Figure 3b, c (b) Light scattering spectra ($\lambda_{sc} = 650$ nm) of a fibrinogen solution with thrombin only (Control) or thrombin with the mixture of origami and two types of aptamers (Ori + T + H: ~ 0.56 nM origami, 20 nM TBA15, 20 nM HD22), aptamers tethered with 5-mer (T-5A-H, 20 nM) or 16-mer polyA linkers (T-16A-H, 20 nM) and by the DNA origami-aptamer nanoarray (Aptarray: ~ 0.56 nM origami, 20 nM TBA15, 20 nM HD22). (c) Relative thrombin activities of different treatments were estimated by the catalytic rate of thrombin with indicated groups. The catalytic rate of thrombin (V_{Cat}) after the indicated treatments was calculated as $V_{Cat} = C_{Fibrinogen} / (t_{1/2} \times C_{Thrombin})$ obtained from the panel b. The concentration of fibrinogen ($C_{Fibrinogen}$) is 1 mg/ml, and the concentration of thrombin ($C_{Thrombin}$) is 12 nM. The data represent the mean \pm s.d.

of three independent experiments.

In view of the Reviewer's comment, to directly study the density (or number) effects of aptamers, we additionally prepared different patterns of bi-aptamer nanoarrays on DNA origami, displaying discrete bi-aptamer numbers with controlled inter-spacings. We have systematically examined thrombin inhibition by the origami nanostructures using clotting reactions monitoring the conversion of fibrinogen into fibrin.

Revised Supplementary Figure S22 Anti-coagulation effects of the Aptarrays with different bi-aptamer density or numbers. (a-b) Schematic representation of Aptarrays with different bi-aptamer patterns. (c-d) Light scattering spectra ($\lambda_{sc} = 650 \text{ nm}$) of a fibrinogen solution with thrombin only (Control) or thrombin with the mixture of origami and two types of aptamers (20 nM TBA15, 20 nM HD22), aptamers tethered with 5-mer (T-5A-H, 20 nM) or 16-mer polyA linkers (T-16A-H, 20 nM) and by Nanostructure V, VI, (Aptarray: $\sim 5 \text{ nM}$ origami, 20 nM TBA15, 20 nM HD22), VII or

IV (Aptarray: ~2.2 nM origami, 20 nM TBA15, 20 nM HD22). (e-f) Relative thrombin activities of different treatments were estimated by the catalytic rate of thrombin with indicated groups. The catalytic rate of thrombin (V_{Cat}) after the indicated treatments was calculated as $V_{\text{Cat}} = C_{\text{Fibrinogen}} / (t_{1/2} \times C_{\text{Thrombin}})$ obtained from the panel c or d. The concentration of fibrinogen is 1 mg/ml, and the concentration of thrombin is 12 nM. The data represent the mean \pm s.d. of three independent experiments.

The results of initial reaction kinetics of thrombin catalysis revealed that four different patterns of nanoarrays with an equivalent bi-aptamers amount (Nanostructure V, VI, VII and IV, containing 20 nM TBA15, 20 nM HD22) exhibited a similar potent inhibition of thrombin (Revised Figure S22). In comparison with T-16A-H, significantly greater inhibitory activities of the clotting reaction (Revised Figure S22) observed from the bi-aptamer nanoarrays might be due to the higher scaffold rigidity of DNA origami nanostructures. The greater electrostatic attraction between the negatively charged DNA template and positively charged surface residues of protein also enhances the interaction of aptamers and thrombin, which may improve the anticoagulant activity for aptarray. Linker rigidity might affect the interaction of bi-aptamer and thrombin (*Nanomedicine: Nanotechnology, Biology and Medicine*, 2012, 8, 673).

These data have been incorporated and discussed in the revised manuscript (Page 7, Line 3), as follows:

...Aptamers tethered with 5-mer or 16-mer polyA linkers (T-5A-H, T-16A-H) induced slower fibrin formation velocities, in comparison with those controlled groups. Thrombin inhibitory performance can be significantly increased ($98.3 \pm 0.2\%$) by arraying an equivalent amount of aptamers in a multivalent form on the DNA origami template with bi-aptamer distances of 5.4 nm (Aptarray: ~ 0.56 nM origami, containing 20 nM TBA15, 20 nM HD22, Figure 3b-c). The Aptarray-induced inhibition is dose-dependent. (Figure S19). These results demonstrated that DNA origami guided bi-aptamer array elicited a potent synergistic effect on thrombin recognition and inhibition. The effect of colocalization induced-multivalency of bi-aptamers is important for thrombin inhibition. For different bi-aptamer arrays with discrete numbers and controlled patterns, the results of the catalytic rate of thrombin revealed that nanoarrays with an equivalent bi-aptamers amount (20 nM TBA15, 20 nM HD22) exhibited a similar potent inhibition of thrombin (Figure S5-7 and S22). In comparison with T-16A-H, significantly greater inhibitory activities of the clotting reaction observed from the bi-aptamer nanoarrays might be due to the higher scaffold rigidity and stronger electronegativity of DNA origami nanostructures^{26,32}.

b) the manuscript doesn't put the results in context of what has already been done with origami and thrombin aptamers nor does it make comparisons with other strategies (it is hard to assess how useful this approach will be) work on type and arrangement of aptamers is largely consistent with published results.

of I would like to see a better survey of the work so far on thrombin decorated DNA / RNA nanostructures and a comparison with results from other approaches, for example using decorated nanoparticles.

Krissanaprasit et al (ref 24, <https://doi.org/10.1002/adma.201808262>) is cited as an example of 'nanostructures with rationally designed geometric features, precise spatial addressability and remarkable biocompatibility' but not as an example of a device designed to inhibit coagulation. Chhabra et al is not cited (<https://doi.org/10.1021/ja072410u>).

Chhabra et al (<https://doi.org/10.1021/ja072410u>) show attachment of thrombin to DNA origami tiles using a row of identical aptamers (given this result it is perhaps surprising that the control tiles in the manuscript don't show any thrombin binding e.g. in the AFM assay shown in Fig S6,7, 8). Can the authors account for this difference? What was the thrombin concentration used in AFM assays.

Rinker (ref 21) looked at the effect of spacing between thrombin aptamers on binding affinity and found a 5.4 nm spacing had a 10-fold higher affinity for thrombin than a 24 nm spacing. This result should be noted because the results in the manuscript on aptamer spacing confirm the results of Rinker.

Similarly, the use of the reverse complement to release thrombin has been described elsewhere - this is a useful property but not a surprising one, the precedent in the literature should be described.

Our response 1b):

We thank the reviewer for bringing up this important issue. In accordance with the reviewer's suggestion, we have cited the mentioned references and introduced these previously reported researches in the revised introduction (Page 4, Line 4).

“...In contrast to traditional nanoparticles, DNA/RNA nanotechnology offers a variety of nanostructures with rationally designed geometric features, precise spatial addressability and remarkable biocompatibility¹⁹⁻²⁴. Several aptamer-tagged DNA/RNA nanostructures have been utilized for immobilizing proteins with defined nanometer spacing and precision²⁴⁻³⁶. Though the reported nanoarchitectures have achieved substantial anticoagulant effects in plasma in test tubes^{24, 26}, in vivo study and clinically relevant validation have yet to be reported. In the current work,....”

The RNA origami bearing multiple thrombin-binding RNA aptamers as anticoagulant reported by Krissanaprasit et al has been introduced and cited as reference 24. in the revised manuscript.

The strategy of aptamer-tagged DNA nanoarchitectures reported by Chhabra et al has

been cited as reference 25 in the revised manuscript.

Distance-dependent multivalent ligand-protein binding strategy introduced by Rinker et al has been cited as reference 23 and discussed in the revised introduction and results part (Page 5, Line 17)

“...The avidity effect of DNA origami-bivalent aptamer assemblies is consistent with the previously reported results^{23, 26}. Therefore, we designed...”

Thrombin binding efficacy is related to protein concentration, or ratio of the aptamer to thrombin. In Chhabra's report (*J. Am. Chem. Soc.*, 2007, 129, 10304), DNA origami with single type of thrombin aptamers (TBA15) can arrange thrombin molecules into the S-shaped pattern. Thrombin molecules (2 μ M) were added to the rectangular-shaped DNA origami containing 19 copies of thrombin aptamers (3 nM origami, containing 57 nM aptamers, ~35:1 ratio of the thrombin to aptamer) for AFM imaging.

Rinker's work (*Nat. Nanotechnol.*, 2008, 3, 418) have demonstrated that thrombin preferred to bind to the dual-aptamer locations on DNA origami templates. In that work, nanoarrays with 6 copies of dual-aptamers (10 nM origami, containing 60 nM dual-aptamers with an inter-molecular distance of ~5.8 nm) were mixed with thrombin molecules (60 nM, 1:1 ratio of the thrombin to dual-aptamer) for AFM imaging.

In our original manuscript, we examined the protein binding of our bi-aptamer nanostructures I-IV (Figure S6-10) at a 1: 1 ratio of the thrombin to the aptamer. Our results are consistent with Rinker's work, especially AFM images of low concentration of thrombin.

Additional AFM imaging (Figure R1) was performed using 0.5 nM TBA15-loaded origami after incubation with different concentrations of thrombin (0, 4.5, 45, 90 nM). When thrombin concentration is increased, the binding efficacy on DNA origami is increased accordingly (Figure R1).

Figure R1 AFM images showing thrombin bound to the aptamer loaded DNA origami at different concentrations.

Minor comments:

2. To many similar ways to describe the same thing leads to confusion (DNA origami nanoarray for anticoagulation, aptarray, DNA nanoanticoagulants, DNA-origami based bi-aptamer arrays). Aptamer on its own called TBA15 / HD22 but, when mixed with origami called T/H, when both aptamers bound aptarray but when only one bound and the other present called TBA15-loaded + HD22 or HD22-loaded + TBA15. Please find a simpler nomenclature so that it is easier to interpret control experiments. Also note that the caption to Fig S18 says aptamer loaded and the figure says aptamer-loading.

Our response 2:

We are grateful to the Reviewer for this insightful comment. In accordance with the reviewer's suggestion, we have used the terms "aptarray" and "DNA origami based bi-aptamer arrays" in the revised manuscript to describe our origami anticoagulant. Aptamer on its own is named TBA15 (T) or HD22 (H). When the aptamers are mixed with origami, it is named Ori + T + H. When only one aptamer bound to the origami and the other is present, the sample is named T-loaded Ori +H or H-loaded Ori +T.

3. Fig. 2.

Fig. 2a (and Fig S13). What temperature was the gel run at? The Alexa 488 channel shows a fast running smear at the bottom of the gel, I would prefer to see the whole gel because I wonder if the smear is evidence of the aptamer dissociating from the origami as the gel is run.

Our response 2:

The agarose gel electrophoresis was performed at room temperature (~ 25 °C). The fast-running smear of Alexa 488 channel of the gel image was evident because partial dye-labelled aptamers dissociated from origami templates during a long electrophoresis process (>1h, 100V). We repeated electrophoresis at 4 °C. The electrophoresis time was shortened to ~30 min and the voltage was decreased to 70V. The updated gel images are now included as the revised **Supplementary Figure S16**

Revised Supplementary Figure 16 Agarose gel images of DNA origami-aptamer nanoarray. Lane 1, M13 genome DNA. Lane 2, bare rectangular DNA origami. Lane 3, Cy5-TBA15 loaded DNA origami. Lane 4, Alexa 488-HD22 loaded origami. Lane 5, Cy5-TBA15 and Alexa 488-HD22 co-loaded origami. The DNA origami co-migrate with Cy5-TBA15 and Alexa 488-HD22, indicating the binding of two types of aptamers to the DNA origami nanostructures.

Fig. 2bc. Why not show both aptarray and origami result in this figure (i.e. move traces from Fig. S15)

Our response 2:

We thank the Reviewer for this suggestion. The combined figures are now presented as the revised Figure 2b and c.

Revised Figure 2b and c (b) Hydrodynamic size of DNA origami template (blue) and Aptarray (red). (c) Zeta potential of bare DNA origami template (blue) and Aptarray (red).

4. Fig. 3bc (and Fig. S17-21 especially Fig. S17). Thrombin activity is described as the initial rate or the linear part of the trace. This requires further explanation. In Fig. S17 for example, I can't see any linear portion of the trace. I think it is worth a more careful consideration of how to interpret these results.

Our response 4:

We thank the Reviewer for this suggestion. According to the previous report (*Angew. Chem. Int. Ed.* **2020**, 59, 17697), the catalytic rate of thrombin (V_{cat}) after the indicated treatments was calculated as $V_{cat} = C_{Fibrinogen} / (t_{1/2} \times C_{Thrombin})$. To clearly show the linear part and the plateau of the trace, the observation time has been extended to 9000s. We have repeated the thrombin inhibition assay and the updated results (**Revised Figure S19-24**) have been included in the revised manuscript. The relative thrombin activity after the indicated treatments was estimated by the catalytic rate of thrombin with indicated groups. The concentration of fibrinogen ($C_{Fibrinogen}$) is 1 mg/ml, and the concentration of thrombin ($C_{Thrombin}$) is 12 nM.

Revised Figure S19 Distance-dependent thrombin inhibition by DNA origami-aptamer assemblies. (a) Schematic drawing of four different DNA nanostructures I-IV (with two rows of 9 of each aptamer arranged ~ 68 nm, 46 nm, 24 nm or 5.4 nm apart). (b) Light scattering spectra ($\lambda_{sc} = 650$ nm) of fibrinogen solution with thrombin only (Control) or thrombin inhibited by aptamers extending origami nanostructures (I-IV) with distances of ~ 68, 46, 24, 5.4 nm. (c) Relative thrombin activities of different treatments were estimated by the catalytic rate of thrombin (V_{Cat}) obtained from the light scattering intensities (b). Data represent the mean \pm s.d. of three independent experiments.

5. In Fig. 3bc, control experiments were done with 20 nM aptamer. Aptarray experiments were done at 'equivalent aptamer amounts' - it says that this is 0.6 nM in the caption but it would be useful to add this to the text. It should also be made clear what the dose-dependent concentrations are in Fig. S19 (I assume that these are aptamer concentrations not aptarray concentrations).

Our response 5:

We thank for the Reviewer for this suggestion. We added the concentration of aptarray in the revised manuscript (Page 7, Line 6).

...Thrombin inhibitory performance can be significantly increased ($98.3 \pm 0.2\%$) by

arraying an equivalent amount of aptamers in a multivalent form on DNA origami template with bi-aptamer distances of 5.4 nm (Aptarray: ~0.56 nM origami, containing 20 nM TBA15, 20 nM HD22, Figure 3b-c)....

We are sorry for the misleading labels in the original Figure S19 (revised Figure S21). Aptarrays used in the dose-dependent thrombin inhibition were ~0.14 nM (containing 5 nM T, 5 nM H), ~0.28 nM (containing 10 nM T, 10 nM H) and ~0.56 nM (containing 20 nM T, 20 nM H).

Reviewer #2 (Remarks to the Author):

This article reports the design and characterization of a DNA-origami platform with anticoagulant activity. Using a DNA origami as a scaffold, the authors were able to spatially introduce arrays of thrombin-binding aptamers positioned at either 68, 46, 24 or 5.4 nm from each other. They found that the 5.4 nm array (distance between two thrombin aptamers TBA and HD22) provides a better binding towards thrombin and therefore displayed a stronger inhibition of thrombin's catalytic activity, an enzyme responsible of coagulation. Using this nanosystem, they demonstrated its anticoagulation properties in human plasma, in whole blood and with mice in vivo study. They further investigated the capacity of using this new nanosystem for extracorporeal hemodialysis while providing technical and safety assessments.

This paper reports an interesting work on a clinically relevant application. Few papers on DNA origami display such a nice clinical validation. The experiments are well done and support the conclusion and the results and figures are very nice.

***My main concern with this paper is that the enhanced effect obtained from this complex origami strategy, can typically be obtain via a much simpler chemical strategy: by chemically attaching together two inhibitors that binds the same target (see e.g. the classic SAR by NMR paper: <https://science.sciencemag.org/content/274/5292/1531>).

This strategy is much simpler and has also already been successfully realized for the TBA-HD22 system employed by the authors: you can find this info on wikipedia:

“Avidity effect of TBA and HD22:

Similar to antibody, aptamers TBA and HD22 show avidity effect against thrombin after dimerization. When TBA and HD22 are conjugated with an optimal linker[19][20] or co-printed on the sensor surface with an optimal density,[21] the affinity against thrombin could be significantly enhanced by 100 to 10,000 fold. Furthermore, the dimerization improves the anticoagulant activity

as well. The TBA-HD22 construct (linked with 16-mer polyA) shows significant improvement both in the assay of activated partial thromboplastin time, clotting time and thrombin-induced platelet-aggregation.”

Unfortunately, the authors fail to mention this fact and did not provide an hypothesis to explain why their origami strategy is working...

If the authors could demonstrate that their origami platform (with its ability to attach aptamers at specific distance from each other) performs better or as good as a simple conjugation strategy (e.g. using the optimal TBA-HD22 conjugate), I may recommend the publication of this article. But if this is not the case, I believe that this strategy remains way too complex, expensive (complex synthesis, assembly and purification) and cumbersome to be useful/commercialized.

Of note, as mentioned by the authors, the origami remains for a longer period in blood due to its bigger size. This effect could also be easily obtained by attaching the small TBA-HD22 conjugate to albumin via the presence of an hydrophobic moiety.

Our response:

We would like to express our gratitude to the Reviewer for this excellent suggestion for additional, previously reported controls. The simple aptamer pair controls considerably strengthen our hypothesis (that two aptamers colocalized on origami elicited a potent synergistic effect on thrombin recognition and inhibition).

We have now repeated the experiments of clotting reactions and added two additional control groups: aptamers were tethered with 5-mer or 16-mer polyA linkers (*ChemBioChem*, 2007, 8, 2223; *Journal of Thrombosis and Haemostasis*, 2008, 6, 2105). We have systematically examined thrombin inhibition by the origami nanostructures using clotting reactions monitoring the conversion of fibrinogen into fibrin.

Revised Figure 3b, c (b) Light scattering spectra ($\lambda_{sc} = 650 \text{ nm}$) of a fibrinogen

solution with thrombin only (Control) or thrombin with the mixture of origami and two types of aptamers (Ori + T + H: ~0.56 nM origami, 20 nM TBA15, 20 nM HD22), aptamers tethered with 5-mer (T-5A-H, 20 nM) or 16-mer polyA linkers (T-16A-H, 20 nM) and by the DNA origami-aptamer nanoarray (Aptarray: ~0.56 nM origami, 20 nM TBA15, 20 nM HD22). (c) Relative thrombin activities of different treatments were estimated by the catalytic rate of thrombin with indicated groups. The catalytic rate of thrombin (V_{cat}) after the indicated treatments was calculated as $V_{\text{cat}} = C_{\text{Fibrinogen}} / (t_{1/2} \times C_{\text{Thrombin}})$ obtained from the panel b. The concentration of fibrinogen ($C_{\text{Fibrinogen}}$) is 1 mg/ml, and the concentration of thrombin (C_{Thrombin}) is 12 nM. The data represent the mean \pm s.d. of three independent experiments.

Through light scattering measurements (revised Figure 3b-c), aptamers tethered with 5-mer or 16-mer polyA linkers (T-5A-H, T-16A-H) resulted in slower fibrin formation velocities, in comparison with the individual free aptamer strands, the mixtures of the aptamers (T + H) or the mixtures of the aptamers and rectangular origami templates at equivalent aptamer amounts (Ori + T + H). The most potent inhibitory performance (98.3 \pm 0.2%) was observed with the addition of the origami-bivalent aptamer nanoarray with distances of ~5.4 nm.

In comparison with T-16A-H, significantly greater inhibitory activities of the clotting reaction observed from the bi-aptamer nanoarrays might be due to the higher scaffold rigidity of DNA origami nanostructures. The greater electrostatic attraction between the negatively charged DNA template and positively charged surface residues of protein also enhances the interaction of aptamers and thrombin, which may improve the anticoagulant activity for aptarray (*Nanomedicine: Nanotechnology, Biology and Medicine*, 2012, 8, 673).

We have now repeated the experiments of *in vivo* anticoagulation and added an additional control group (T-16A-H). The updated results have been included in the revised manuscript (**Revised Figure 4**). We assessed the *in vivo* thrombin-inhibiting activity of the T-16A-H and aptarray by APTT. Mice were treated with buffer, T-16A-H or aptarray *via* a single tail vein injection. For neutralization study *in vivo*, DNA antidotes were administrated to mice followed by aptarray. Plasma was collected from treated mice and APTT was measured. The aptarray-treated mouse group exhibited the longer clotting time (41.7 \pm 2.7s) post intravenous injection in comparison with buffer treated (19.4 \pm 0.7s) and T-16A-H treated group (29.7 \pm 1.8s). Consistent with the plasma assay, neutralization by antidote solution through tail injection demonstrated that DNA antidote rapidly and effectively neutralized the anticoagulant effect of aptarray.

The *in vivo* anticoagulation and neutralization ability were also evaluated by using murine tail-transection bleeding models. After the mice received the origami-based anticoagulant through intravenous injection, the tails of the mice were clipped and the blood lost from the tail over the next 15 min was collected and determined. Mice treated

with aptarray exhibited obvious hemorrhagic effect in response to the trauma, in contrast to buffer treated and T-16A-H treated group. Administration of DNA antidote in mice completely prevented the excessive bleeding induced by the origami-based anticoagulant and surgical trauma. The APTT and tail-transection bleeding results indicated that the increased hydrodynamic sizes and stability by the origami template can prolong the circulation time of aptarray, which induce more efficient anticoagulation compared with free bi-aptamers.

Revised Figure 4 Anticoagulation and the neutralization studies in mice. (a) Mice ($n = 5$) were treated with buffer (100 μL), T-16A-H (20 μM , 100 μL) or aptarray (~ 560 nM, 100 μL) *via* a single tail vein injection. For the neutralization study *in vivo*, DNA antidotes were administrated intravenously to mice followed by aptarray. Plasma was collected from treated mice and APTT was measured. (b) Schematic drawing of a murine tail-transection bleeding model. (c) Mice ($n = 5$) were treated with buffer (100 μL), T-16A-H (20 μM , 100 μL) or aptarray (~ 560 nM, 100 μL). DNA antidotes were then administrated intravenously to the anticoagulated animals for neutralization. The tail tips were amputated and the blood loss of mice was measured. Data represent the mean \pm s.d. Statistical significance was calculated by one-way ANOVA with the Tukey post hoc test. NS, $P > 0.05$; ** $P < 0.01$; **** $P < 0.0001$.

These data have been incorporated and discussed in the revised manuscript (**revised Figure 4**).

Dual aptamer conjugated albumin for enhanced thrombin binding and inhibition has not been reported yet. It will not be easy to site-specifically anchor two DNA aptamers on albumin and control the distance and orientation of the two attaching aptamers, which will affect the thrombin inhibitory effects.

The DNA origami technique was introduced by Rothemund in 2006. A desired DNA origami structure can be constructed by a long scaffold ssDNA molecule (usually M13 bacteria phage genome DNA) folded into an arbitrary architecture using hundreds of short single strands (staple strands) that fix the scaffold's conformation. There are several approaches can be utilized for scalable production of staple and scaffold strands, such as the use of bacteriophage-infected *E. coli* for scaffold (*Proc. Natl. Acad. Sci.*, 2007, 104, 6644; *Nano Lett.*, 2015, 15, 4672) and the "monoclonal stoichiometric"

(MOSIC) method for short ssDNA (*Nat. Methods*, 2013, 10, 647). With the advances in biotechnological production of scaffolds and staples, the cost of folded DNA origami has been greatly reduced from about US\$200 per milligram to around 20 cents (*Nature*, 2017, 552, 84; *Nature*, 2017, 552, 34). Litre-scale ssDNA production (M13 scaffold) has been achieved in our lab through fermentation of phage-infected *E. coli*. Thus far, hundreds of milligram-scale production of DNA origami nanostructures has been realized.

Figure R3 Photograph of lyophilized DNA origami nanostructures.

These data have been incorporated and discussed in the revised manuscript (Page 13, Line 13), as follows:

....the coagulation cascade. With the advances in mass production strategies of DNA nanostructures³⁵, the cost of origami-based anticoagulant can be greatly reduced, allowing for future clinical study. Our potent,...

Here are other suggestions that I think may improve this paper:

1) Multiple typos are found throughout the manuscript. This includes missing coma, inverted letters (ex. apatmers vs aptamers), missing words (ex. “which is consistent with the aptamer’s partial inhibitory effects”), double wording (ex. “In contrast to from traditional nanoparticles”, missing letter. A closer look at typos should be done prior to submission.

Our response 1:

We are sorry for the mistakes in the text. We accordingly made corrections in the revised manuscript.

2) On page 5 and 6, authors mentioned briefly why the DNA origami with 5.4 nm between each aptamer provides a more efficient binding with thrombin compared to DNA origami with distances of 64, 46 and 24 nm. They mentioned a potent synergistic effect. Authors should develop more on the chemical nature of that proposed synergistic effect and mentioned information about the distance between the two exosites and how this can enable both aptamers to bind simultaneously the thrombin therefore increasing the affinity of thrombin for the Aptarray. As

mentioned above, please cite the literature about the “Avidity effect” (how to obtain a “better” inhibitor with smaller K_d) by attaching two recognition elements to each other...

Our response 2:

TBA 15 is a 15-base-long, single-stranded DNA oligonucleotide that can directly bind to exosite I of thrombin (K_d ~70-100 nM) and elicit a potent anticoagulant activity. Another anti-thrombin aptamer, HD22, recognizes thrombin’s exosite II with a high binding affinity (K_d ~ 0.5 nM). Both aptamer sequences are known to have a stem/binding region, using only a few bases for attaching with the surface of thrombin. In order to find an optimal relative inter-spacing of the two aptamers, nanostructures I-IV (with two rows of 9 of each aptamer arranged ~ 68 nm, 46 nm, 24 nm or 5.4 nm apart) were designed. Thrombin binding efficiency was determined from AFM images (Figure S6-9, containing > 150 origami structures) by dividing the number of thrombin-binding structures (highlighted by the red circle) by the total number of origami assemblies counted. The efficiencies of thrombin loading of nanostructure I (inter-aptamer distance ~ 68 nm), II (inter-aptamer distance ~ 46 nm) and III (inter-aptamer distance ~ 24 nm) were less than 10%. Approximately 76% of the nanostructure IV contained thrombin molecules on the surface (Figure S10). As the inter-aptamer distance of 5.4 nm matches the dimension of thrombin molecule (~4 nm), the two aptamers can act as a bivalent single molecular species that display a stronger binding affinity to the protein than any one of the individual aptamers does alone.

It is reported that the bivalent binding of aptamers placed at an optimized distance can have a binding affinity (K_d ~ 0.1 nM) better than the values for the monovalent binding arrangements (*Nat. Nanotechnol.* 2008, 3, 418-422).

According to the Reviewer’ comments, the literature about the avidity effect have been cited in the revised manuscript (Page 5, Line 15), as follows:

...with distances of 5.4 nm (Figure S9-13). As the inter-aptamer distance of 5.4 nm matches the dimension of thrombin molecule (~4 nm), the two aptamers can act as a bivalent single molecular species that display a stronger binding affinity to the protein than any one of the individual aptamers does alone. The avidity effect of DNA origami-bivalent aptamer assemblies is consistent with the previously reported results^{23, 26}. Therefore, we designed....

3) On page 6, authors measured the size and the zeta potentials of their DNA origamis and provided some values. The authors should explain why they measured the zeta potential and the importance of this measure to characterise their two assemblies.

Our response 3:

The size and the zeta potentials of nanoparticles are important features for their behaviors in vitro and in vivo. For instance, in the dialysis circuit, a possible limitation

of thrombin-binding aptamers is their high diffusion rate through the dialyzer due to their relatively low molecular weights (<10 kDa). Most proteins in the bloodstream are negatively charged (*Nat. Nanotechnol.*, 2013, 8, 772), which may absorb positively charged nanoparticles and affect their interaction with biological systems.

Aptarrays can be constructed with suitable sizes and hydrophilic surfaces which may prolong the circulation time of the dialyzer. DNA-based aptarray with the negatively charged surface may have fewer interactions with proteins in the bloodstream compared with positively charged nanoparticles. Before the coagulation studies, we tested the size and zeta potentials of Aptarray. Then we assessed whether Aptarrays can show increased retaining time in the circuit and better anticoagulant properties.

We added the brief explanation for providing characteristic results in the revised manuscript (Page 6, Line 6).

.... respectively. The size and the zeta potentials of nanoparticles are important parameters to affect their behaviors in vitro and in vivo. Dynamic light scattering (DLS) analysis...

4) On page 7, authors mentioned two specific measurements to quantify anticoagulation properties. These two measurements are called APTT (Activated Partial Thromboplastin Time) and PT (Prothombin Time). A better explanation of these two values should be done in order to better appreciate the results. Authors should explain these two parameters the same way they explained the concept of thromboelastography (TEG) on page 8 (Fig. S24).

Our response 4:

We thank the Reviewer for this suggestion. We added a brief description of APTT/PT in the revised manuscript (Page 8, Line 10)

...using activated partial thromboplastin time (APTT; Figures 3d and S22) and prothrombin time (PT; Figure S22). The APTT and PT are commonly used medical tests that assess a person's blood coagulation process and monitor anticoagulation therapy. These assays evaluate the amount and the function of clotting factors that are important for blood clot formation.³³ The APTT examines the activity of the intrinsic and common clotting pathways, while the PT assay measures the integrity of the extrinsic and common systems of coagulation....

5) On figure S12, we don't see clearly the band of the redundant aptamers strand that are eliminated by PEG-induced precipitation. A better light contrast should resolve that problem. On figure S17, the x axis is missing. On the DNA sequence at the end of the SI, authors must mention the 3' and 5' extremities.

Our response 5:

We thank the reviewer for bringing these points to our attention.

We have now repeated the gel electrophoresis. The updated results are included in the revised **Supplementary Figure 15** in the revised manuscript.

Revised **Supplementary Figure 15c** (c) Agarose gel image of unpurified (left) and PEG-purified Aptarray.

We have labeled the x-axis as Time (sec) in the original **Supplementary Figure 17** (revised **Supplementary Figure 19**) in the revised submission. DNA sequences used in our experiments are now included in Supplementary Note. All the sequences are present from 5'-3' (left to right).

6) Throughout the manuscript, authors use different names to design the Aptarray. Sometimes it is referred as DNA nanoanticoagulant, DNA origami-aptamer nanoarray, origami-based anticoagulant, DNA origami-based aptamer nanoarray, DNA origami-based anticoagulants, DNA origami-based nanoagent. Several terms for the same thing only create confusion therefore one term should be prioritized to describe the Aptarray.

Our response 6:

We are grateful to the Reviewer for this insightful comment. In accordance with the reviewer's suggestion, we have used the terms "aptarray" and "DNA origami based bi-aptamer arrays" in the revised manuscript to describe our origami anticoagulant.

REVIEWERS' COMMENTS

Reviewer #1 (Remarks to the Author):

The authors have made significant improvements to the manuscript in response to the reviewer comments, I would be happy to see the manuscript published in its present form.

Like any good manuscript it opens up new questions that must be answered - why is origami-aptamer array effective (what role does shape and charge of the origami have to play), there are also further interesting questions to ask about number and density of the aptamers

Reviewer #2 (Remarks to the Author):

The authors have demonstrated that their origami platform (with its ability to attach aptamers at specific distance from each other) performs better than a simple conjugation strategy (e.g. using the optimal TBA-HD22 conjugate). Very nice result.

They have also constructively responded to all other comments that I had. I recommend the publication of this article.

Best regards, Alexis Vallée-Bélisle

**RE: Final revisions for Nature Communications manuscript
NCOMMS-20-08022B-Z**

Point-to-point response to Reviewer

Reviewer #1 (Remarks to the Author):

The authors have made significant improvements to the manuscript in response to the reviewer comments, I would be happy to see the manuscript published in its present form.

Like any good manuscript it opens up new questions that must be answered - why is origami-aptamer array effective (what role does shape and charge of the origami have to play), there are also further interesting questions to ask about number and density of the aptamers

Our response:

We are grateful for the reviewer's positive comment.

Our results (Figure 3b-c) demonstrated that origami-aptamer array elicited a more effective thrombin recognition and inhibition, in comparison with aptamers tethered with 5-mer or 16-mer polyA linkers (T-5A-H, T-16A-H). The results indicated that precise assembly induced-multivalency of aptamers is important for thrombin inhibition. In the mouse model, our results (Figure 4) indicated that the increased hydrodynamic sizes by the origami template can prolong the circulation time of Aptarray, which induce more efficient anticoagulation compared with T-16A-H *in vivo*.

The more potent inhibitory activities of origami-aptamer array might be due to the higher scaffold rigidity of rectangular DNA origami nanostructures that provide more control of the orientation of bi-aptamer pairs, compared with T-5A-H/T-16A-H. The greater electrostatic attraction between the negatively charged DNA origami template and positively charged surface residues of protein also enhances the interaction of aptamers and thrombin, which may improve the anticoagulant activity for origami-aptamer array.

For nanoarrays with different numbers and densities of the aptamers, our results (Figure S5-7 and S22) revealed that nanoarrays with an equivalent bi-aptamers amount exhibited a similar potent thrombin inhibition. Further investigation of the shape effect of DNA template and the role of aptamer number/density on thrombin inhibition will be performed in the future study.

We have included the following discussion in the revised manuscript (Page 13, Line 8):

...of the aptamers. The increased hydrodynamic sizes by the origami template can prolong the circulation time of Aptarray, which induce more efficient anticoagulatory

effects compared with free bi-aptamers tethered with polyA linkers *in vivo*. In an *ex vivo*...

Reviewer #2 (Remarks to the Author):

The authors have demonstrated that their origami platform (with its ability to attach aptamers at specific distance from each other) performs better than a simple conjugation strategy (e.g. using the optimal TBA-HD22 conjugate). Very nice result.

They have also constructively responded to all other comments that I had. I recommend the publication of this article.

Our response:

We are very grateful for the reviewer's positive comments.